# Lake mixing regime selects apparent methane-oxidation kinetics of the methanotroph assemblage

Magdalena J. Mayr[1,2]*, Matthias Zimmermann[1,2]*, Jason Dey[1], Bernhard Wehrli[1,2] and Helmut Bürgmann[2]

* These authors contributed equally to this work.

[1]Department of Surface Waters—Research and Management, Eawag, Swiss Federal Institute of Aquatic Science and Technology, Kastanienbaum, Switzerland

[2]Institute of Biogeochemistry and Pollutant Dynamics, Department of Environmental Systems Science, ETH Zurich, Swiss Federal Institute of Technology, Zurich, Switzerland

*Correspondence to*: Magdalena J. Mayr (magdalena.mayr@eawag.ch)

**Abstract.** In lakes, large amounts of methane are produced in anoxic sediments. Methane-oxidizing bacteria effectively convert this potent greenhouse gas into biomass and carbon dioxide. These bacteria are present throughout the water column where methane concentrations can range from nanomolar to millimolar concentrations. In this study, we tested the hypothesis that methanotroph assemblages in a seasonally stratified freshwater lake are adapted to the contrasting methane concentrations in the epi- and hypolimnion. We further hypothesized that lake overturn would change the apparent methane oxidation kinetics as more methane becomes available in the epilimnion. Together with the change of methane oxidation kinetics, we investigated changes in the transcription of genes encoding methane monooxygenase, the enzyme responsible for the first step of methane oxidation, with metatranscriptomics. We show in laboratory incubations of the natural microbial communities that the half-saturation constant ($K_m$) for methane – the methane concentration at which half the maximum methane oxidation rate is reached – was 20 times higher in the hypolimnion than in the epilimnion during stable stratification. During lake overturn, however, the kinetic constants in the epi- and hypolimnion converged along with a change of the transcriptionally active methanotroph assemblage. Conventional particulate methane monooxygenase appeared to be responsible for methane oxidation under different methane concentrations. Our results suggest that methane availability is one important factor for creating niches for methanotroph assemblages with well-adapted methane-oxidation kinetics. This rapid selection and succession of adapted lacustrine methanotroph assemblages allowed the previously reported high removal efficiency of methane transported to the epilimnion to be maintained even under rapidly changing conditions during lake overturn. Consequently, only a small fraction of methane stored in the anoxic hypolimnion is emitted to the atmosphere.

## 1 Introduction

Lakes are an important source of greenhouse gases. Methane is a major contributor to the climate impact of the greenhouse gas emissions from lakes (DelSontro et al., 2018). The oxidation of the strong greenhouse gas methane in freshwater lakes is mainly achieved by methane-oxidizing bacteria (MOB), which have the unique ability to use methane as their sole carbon and energy source (Hanson and Hanson, 1996). In seasonally stratified lakes, large amounts of methane, which is produced as a final product of anaerobic organic matter degradation, can accumulate in the oxygen-depleted hypolimnion (Conrad, 2009; Steinsberger et al., 2017). Under stratified conditions, aerobic and sometimes anaerobic MOB oxidize this methane in the water column, thereby preventing diffusive outgassing (Bastviken et al., 2002; Graf et al., 2018; Mayr et al., 2020a). Lake overturn in autumn leads to mixing of the oxygen-rich surface layer with the methane-rich bottom water (Schubert et al., 2012). The simultaneous availability of oxygen and methane promotes growth of MOB in the expanding epilimnion at the surface (Kankaala et al., 2007; Mayr et al., 2020b; Schubert et al., 2012; Zimmermann et al., 2019). The resulting increase in methane oxidation capacity has been shown to be associated with a shift in the MOB assemblage in the epilimnion, which grows fast enough to prevent 90% of the methane transported into the epilimnion from escaping to the atmosphere (Mayr et al., 2020b; Zimmermann et al., 2019).

In temperate, seasonally stratified lakes, the diverse MOB assemblage shows a clear vertical structure and succession during autumn overturn (Kojima et al., 2009; Mayr et al., 2020b). This suggests that mechanisms of spatial and temporal niche partitioning maintain diversity within this functional group (Mayr et al., 2020a). The differences in the methane and oxygen availability in the two water bodies above and below the oxycline likely place very different demands on the ecophysiology of the resident MOB assemblages. Although previous studies have shown great diversity and adaptability of methane oxidation kinetics (Baani and Liesack, 2008; Dunfield and Conrad, 2000; Lofton et al., 2014; Tveit et al., 2019), the role of different kinetic parameters in rapidly changing lake environments has so far not been studied systematically. Here, we hypothesized that apparent kinetic parameters of methane oxidation vary between epi- and hypolimnion and that kinetic parameters vary seasonally together with the MOB assemblage, which would show that methane availability is a driver of apparent methane oxidation kinetics of the MOB assemblage. Further, the methane affinity of lacustrine MOB especially in the epilimnion has implications for the amount of methane outgassing during both, stable stratification and lake overturn.

The first step of methane oxidation is mediated by the methane monooxygenase. Most MOB possess the copper-dependent particulate form of the methane monooxygenase (pMMO). Known isozymes of pMMO have been shown to exhibit different methane oxidation kinetics, including high affinity variants that are able to oxidize methane even at atmospheric concentrations (Baani and Liesack, 2008; Dam et al., 2012). A subset of MOB encode the soluble MMO (sMMO) that has a lower methane affinity than pMMO and has been hypothesized to be used by MOB under high methane concentration, because MOB biomass is assumed to be higher under such conditions leading to copper limitation and a switch to copper-free sMMO (Semrau et al., 2018). The abundance of the sMMO gene has been found to be low in Lake Rotsee (Guggenheim et al., 2019), but relative transcription between epi- and hypolimnion has not been investigated so far.

In this study we conducted a combined kinetic and metatranscriptomic analysis in a small pre-alpine lake to test our hypothesis that MOB assemblages show distinct apparent methane oxidation kinetics in the methane-rich hypolimnion compared to the epilimnion with low methane concentrations. Further, we examined the changes in apparent methane oxidation kinetics over time during lake overturn, as more methane becomes available in the epilimnion. To do so, we used laboratory incubations of the resident microbial community to measure methane-oxidation rates and methane affinity combined with MOB cell counts. In parallel, we applied metatranscriptomics to characterize the MOB assemblage as well as genes and transcripts involved in the methane oxidation pathway, aiming to link observed changes in apparent methane oxidation kinetics with changes in the MOB population activity. Knowledge about the variability of kinetic parameters of methane oxidation is important to better understand the ecology and physiology of MOB in the environment. Further, our results will inform trait-based or process-based modelling approaches, because a single set of time and space invariant kinetic parameters may not reflect natural conditions adequately.

## 2 Methods

### 2.1 Study site and physicochemical lake profiling

Lake Rotsee is a small eutrophic lake in central Switzerland that is 2.5 km long, 200 m wide and has a maximum depth of 16 m. For more details see Schubert et al. (2012). We profiled and sampled the water column during four campaigns in autumn 2017 at the deepest point of Lake Rotsee at 47.072 N and 8.319 E. We measured profiles of temperature and pressure (depth) with a CTD (RBRmaestro, RBR, Canada). A micro-optode (NTH-PSt1, PreSens, Germany) attached to the CTD measured profiles of oxygen concentrations.

### 2.2 $^3$H-CH$_4$ tracer technique

We used the radio $^3$H-CH$_4$ tracer technique as described in Bussmann et al. (2015) and Steinle et al. (2015) to measure apparent methane oxidation rates and kinetics of the MOB assemblage above and below the oxycline. Similar measurements have been done by Lofton et al. (2014), who derived apparent methane oxidation kinetics from methane oxidation rates using $^{14}$C-CH$_4$. We used the $^3$H-CH$_4$ tracer technique because it is more sensitive than the $^{14}$C-CH$_4$ technique and therefore allows shorter incubation times and rate determination at low CH$_4$ concentrations. We added 200 µL of gaseous $^3$H-CH$_4$/N$_2$ mixture (~80 kBq, American Radiolabeled Chemicals, USA). The specific activity of $^3$H-CH$_4$ is 0.74 TBq mmol$^{-1}$ and the 200 µL of gaseous $^3$H-CH$_4$/N$_2$ mixture therefore contained 108 pmol $^3$H-CH$_4$. In comparison, the 500 µL gas bubble with the lowest concentration of unlabelled methane (see section 2.3), contained 17 nmol CH$_4$. We measured total and water fraction radioactivity in a liquid scintillation counter (Tri-Carb 1600CA, Packard, USA) by adding 1 mL sample to 5 mL Insta-Gel (PerkinElmer, Germany). From these activities, we calculated the methane oxidation rate ($r_{MOx}$):

$$r_{MOx} = [CH_4] \times \frac{A_{H_2O}}{A_{H_2O} + A_{CH_4}} \times \frac{1}{t}$$

where t is time, [CH$_4$] is the concentration of methane and activities (A) were corrected for fractional turnover in killed controls.

### 2.3 Apparent methane oxidation kinetics of microbial communities

We assumed that the dependence of the methane oxidation rate ($r_{MOx}$) of the microbial community on the methane concentration can be described by a Monod kinetics:

$$r_{MOx} = V_{max} \frac{[CH_4]}{K_M + [CH_4]}$$

where $V_{max}$ is the maximum methane oxidation rate and $K_M$ is the half-saturation constant for methane. We use the term *affinity* as the inverse of the half-saturation constant: $1/K_m$. We determined the two kinetic parameters in laboratory incubations of water samples from above and below the oxycline. We collected water from the two depths in 2 L Schott bottles and transported them to the lab dark and cooled. We stripped dissolved methane by bubbling air for 1h. This also removed H$_2$S from the sample which would otherwise reduce the sensitivity of the $^3$H-CH$_4$ technique. For each depth, we prepared 60 mL incubations with 10 different methane concentrations and a killed control. We prepared incubations and controls in triplicates except for the first field campaign where we only prepared duplicates. By adding a 500 µL gas bubble from pre-diluted gas stocks we established methane concentrations of 0.4 to 60 µM. Gas stocks were prepared by evacuating and flushing 120 mL crimp-sealed serum vials with pure nitrogen gas five times and adding defined volumes of methane gas with gas tight syringes. The killed controls were treated in the exact same way as the samples with the exception that we inhibited methane oxidation by adding 1 mL of ZnCl$_2$ (50 % w/v). To start the incubations, we added the $^3$H-CH$_4$ tracer as described in the above section. After vigorous shaking for 1 minute, we kept the incubations dark in a shaker with 100 RPM. We incubated both samples from above and below the oxycline at the temperature measured within the oxycline. While the determined kinetic parameters may thus differ from in-situ values, this approach allows for a direct comparison of the two datasets. After 4 hours, we stopped the incubations by adding 1 mL of ZnCl$_2$ (50 % w/v). We determined the methane oxidation rate in each incubation as described above. Except for the first sampling date, we measured each incubation replicate twice. This resulted in 594 measurements, 72 single

measurements and 261 measurement duplicates. We averaged measurement replicates resulting in 333 data points.

We used a non-linear least squares Levenberg-Marquardt algorithm to fit the Monod equation to the data. Outliers in the data were removed using the following criteria: For the replicates of each methane concentration we removed data points (1) with a water fraction radioactivity that was outside $2\sigma$ from the average water fraction radioactivity of all replicates, (2) which showed a water fraction radioactivity that was not above $2\sigma$ from the background water fraction radioactivity, (3) for which we had less than two replicates after the removal of outliers, (4) with a resulting methane oxidation rate outside $2\sigma$ from the average methane oxidation rate of all replicates, and (5) showing a methane oxidation rate that was higher than the methane oxidation rate measured for the replicates with the highest methane concentration. The $2\sigma$ approach is one recommended approach for outlier detection (e.g. Leys et al., 2013). Because we only had incubation duplicates for the first sampling date, it was not possible to detect outliers based on $2\sigma$ for this campaign and we kept both values in the analysis. The average water fraction radioactivity of the killed controls was used as background radioactivity in the outlier detection procedure. In total 221 datapoints were finally considered in the analysis (66% of all datapoints without measurement replicates). For the five individual outlier criteria, the percentages of detected outliers are: (1) 4%, (2) 4%, (3) 3%, (4) 19%, (5) 3%. The high percentage of outliers for criteria 4 is related to the fact that methane oxidation rates are associated with a higher error than individual measurements because they are computed from multiple individual measurements.

The base value of the *specific affinity* a° is defined as the ratio $V_{max}/K_M$. We approximated mean and variance of the ratio of the two random variables with known mean and variance using the Taylor expansions given in Stuart and Ord (2009).

## 2.4 Methane oxidation rates of the microbial community

We determined the methane oxidation rate of the natural microbial community in duplicate laboratory incubations of water samples from above and below the oxycline. We anaerobically filled water into 60 mL serum vials, and crimp-sealed and transported them to the lab dark and cooled. For each depth, we prepared killed controls with 1 mL of $ZnCl_2$ (50 % w/v) in duplicates in the same way. We started the incubations by adding the $^3H$-$CH_4$ tracer as described above. After vigorous shaking for 1 minute, we kept the incubations dark in a shaker with 100 RPM at the temperature measured within the oxycline. After 4 hours, we stopped the incubations by adding 1 mL of $ZnCl_2$ (50 % w/v).

## 2.5 Methane concentration measurement

We measured *in-situ* methane concentrations in the water column using the headspace equilibration method. For each depth, we collected water samples in 120 mL crimp-sealed serum vials with a small amount of $CuCl_2$ to stop biological activity. We measured methane concentrations in the headspace with a gas chromatograph (Agilent 6890N, USA) equipped with a Carboxen 1010 column (Supelco 10 m × 0.53 mm, USA) and flame ionisation detector. Samples that exceeded the calibration range were diluted with $N_2$ and measured again. We calculated dissolved methane concentrations according to Wiesenburg and Guinasso (1979).

## 2.6 Quantification of methanotroph cells

We investigated the abundance of aerobic methanotrophs by catalysed reporter deposition fluorescence in situ hybridisation after Pernthaler et al. (2002). We fixed water samples of 5 mL with 300 µl sterile filtered (0.2 µm) formaldehyde (2.22% v/v final concentration) for 3 – 6 h on ice. We filtered the samples onto 0.2 µm nuclepore track-etched polycarbonate membrane filters (Whatman, UK), that we dried, and stored at -20 °C until further analysis. We permeabilized cells with lysozyme (10 mg mL$^{-1}$) at 37 °C for 70 min, and inactivated endogenous peroxidases with 10 mM HCl for 10 min at room temperature. To hybridise the filters, we used a hybridisation buffer (Eller et al., 2001) containing HRP-labelled probes at 46 °C for 2.5 h. Furthermore, the buffer contained either a 1:1:1 mix of Mg84, Mg705, and Mg669 probes targeting methanotrophic *Gammaproteobacteria* or a Ma450 probe targeting methanotrophic *Alphaproteobacteria* (Eller et al., 2001). To amplify the fluorescent signals, we used the green-fluorescent Oregon Green 488 tyramide (OG) fluorochrome (1 µl mL$^{-1}$) at 37 °C for

30 min. We counterstained hybridised cells with DAPI (20 µl of 1 µg mL$^{-1}$ per filter) for 5 minutes. For microscopy, we used a 4:1 mix of Citifluor AF1 (Electron Microscopy Sciences, Hatfield, PA, USA) and Vectashield (Vector, Burlingame, CA, USA) as mountant. We used an inverted light microscope (Leica DMI6000 B, Germany) at a 1000-fold magnification to quantify MOB cell numbers. For each sample, we took 22 image pairs (DAPI and OG filters) of randomly selected fields of view (FOVs). To detect and count cells we used digital microbial image analysis software Daime 2.0 (Daims, 2009).

## 2.7 Metagenome and metatranscriptome analysis

We collected lake water with a Niskin bottle and filtered 800 – 2300ml on-site onto 0.2 µm pore size GTTP isopore filters (Merck Millipore Ltd.). To keep the filtration time as short as possible (typically <10 min) and at the same time retrieve enough RNA for sequencing, a 142 mm diameter filter was used. To minimize sample perturbation the filtration device was connected directly to the Niskin bottle. The filters were preserved immediately on dry ice and stored at -80 °C until extraction. We did not apply prefiltration, because filamentous methanotrophs can reach lengths of >100 µm (Oswald et al., 2017). We extracted DNA and RNA with the Allprep DNA/RNA Mini Kit (Qiagen) and treated RNA with the rigorous option using the Turbo DNA-free kit (Invitrogen) to remove remaining DNA. To increase the confidence in the measurement a second filter of the January hypolimnion sample was extracted and sequenced separately. This replicate is shown as Jan (r). RNA yields from the October sampling were deemed insufficient for sequencing as no typical RNA bands were visible during quality control and therefore these samples were omitted from metatransciptome analysis. Metagenomic and metatranscriptomic 150bp paired-end sequencing was done on a NovaSeq 6000 sequencer (Illumina) at Novogene (HK) company limited (Hong Kong, China). Ribosomal RNA was depleted with Ribo-Zero Magnetic Kit (Illumina) prior to sequencing. The co-assembly of metagenomic sequences alone yielded less *pmoA* as well as *pmoB* and *pmoC* sequences than expected, likely due to low coverage. Therefore, we combined predicted genes from both the metagenomic and the metatranscriptomic *de-novo* assembly as described below. Due to low coverage of *pmoA*, *pmoB* and *pmoC* in the metagenome, we used the metagenome only in the assembly process. All further analyses relied on the metatranscriptome.

We removed remaining ribosomal sequences from metatranscriptomic reads with sortmerna v2.1 (Kopylova et al., 2012) and performed quality filtering with trimmomatic v0.35 (Bolger et al., 2014), resulting in 26.6 million - 34 million high quality reads. We co-assembled reads from seven metatranscriptomic libraries using megahit v1.1.3 (Li et al., 2015) with a final k-mer size of 141 and a minimum contig length of 200. This resulted in 2166829 contigs with an average of 672bp and a N50 of 733bp. For quality filtering of metagenomic reads we used prinseq-lite v0.20.4 (Schmieder and Edwards, 2011) with dust filter (30) and a quality mean of 20, resulting in 31.1 million – 37.1 million high quality reads. Again, we performed a co-assembly using megahit of 10 metagenomes (including three October samples without corresponding metatranscriptome) with a final k-mer size of 121 and a minimum contig size of 300bp (4237394 contigs, average 1008bp and N50 of 1250bp). We measured one additional depth in October in between epi- and hypolimnion, which is included in the data repository but is not discussed here. We did not pursue the intermediate sample in later campaigns since we focused our effort on the epilimnion and hypolimnion (continuing with measurement triplicates). We did however use the metagenomics data from this sample for the assembly. Gene prediction for both co-assemblies was done with prodigal v 2.6.3 (setting: meta, Hyatt et al., 2010). After combining the predicted genes, cd-hit-est v4.6.6 (Li and Godzik, 2006) was used to remove very similar and duplicate (identity 0.99) predicted genes. With Seqkit v0.7.2 (Shen et al., 2016) predicted genes shorter than 400bp were removed. Predicted genes encoding pMMO were annotated with prokka v1.3 (Seemann, 2014) using the incorporated databases (metagenome option) and diamond blastx v0.9.22 (e-value 10$^{-6}$, Buchfink et al., 2014) against custom databases for *pmoA*, *pmoB* and *pmoC*. These custom databases included *pxmABC*, *pmoCAB2*, *pmoCAB* from both alpha- and gammaproteobacterial genomes, which were extracted manually. The databases are provided as supplementary files 1-3. Annotation was manually validated using alignments and the NCBI refseq_protein database (22.4.2019, O'Leary et al., 2016). *pmoA, pmoB and pmoC* variants summing to a cross-sample sum higher than 50 transcripts per million (TPM) were retained. Genes annotated as *pmoA, pmoB and pmoC* variants which were either not the expected gene (manual inspection) or shorter than 400 bp were removed. Genes encoding part of the soluble methane monooxygenase sMMO (*mmoX, mmoY and mmoZ*) were annotated with prokka v1.3 using incorporated databases and the metagenome option. Paired-end metatranscriptomic reads were mapped to the predicted genes

using bbmap v35.85 (Bushnell, 2014) at an identity of 0.99 without mapping of ambiguous reads, and then converted with samtools v1.9 (Li et al., 2009) and counted with featurecounts (Liao et al., 2014) of subread v1.6.4 package (-p option). The count table was normalized within samples to transcripts per million (TPM, Wagner et al., 2012) by first dividing the counts by gene length, then the result by gene was divided by the sum of all results times one million. The TPM values (suppl. file 4) were used to produce the figures in R. The correspondence analysis (CA) was performed with vegan (v2.5.6, Oksanen et al., 2019) in R (v3.5.2, R Core Team, 2018) based on the combined and square root transformed TPM values of *pmoCAB* data (scaling=2).

All sequences were classified to the family level based on the NCBI refseq_protein database using blastx (O'Leary et al., 2016). Further classification was based on a *pmoA* phylogenetic tree (shown in the supplementary material). *pmoA* amino acid sequences were derived with MEGA7 and aligned with Muscle (Kumar et al., 2016). A neighbor-joining tree was inferred using 10000 bootstrap replications with Poisson correction method based on 131 positions. Known cultivated or uncultivated groups were assigned at bootstrap values >0.7 and a protein similarity >94%, corresponding to genus level resolution according to Knief (2015). The *pmoB* and *pmoC* sequences were assigned to these groups if originating from the same contig. For many sequences a more detailed taxonomic assignment than family was not possible, and therefore labelled "unclassified type Ia" and "unclassified type II", respectively.

## 3 Results and discussion

### 3.1 Environmental conditions during the autumn overturn

Lakes located in climatic zones with strong seasonal variability, show seasonal vertical stratification of their water masses that is fundamental for all physical, chemical and biological processes occurring within them (Boehrer and Schultze, 2008). During the warm season, the increasing temperature at the lake surface establishes two physically and chemically different water masses in the lake, the epilimnion and the hypolimnion. The epilimnion at the surface is warmer, well-mixed and has continuous supply of oxygen from the atmosphere and photosynthesis. In contrast, the colder and denser hypolimnion is physically separated from the epilimnion and generally shows diffusive gradients of dissolved substances. During the cold season, surface cooling leads to vertical mixing which gradually deepens the well-mixed surface layer and mixes hypolimnetic water into the surface layer. During this autumnal overturn period, both temperature and chemistry of the surface water change and potentially create new ecological niches. In the following we label the water masses above and below the thermocline as 'epilimnion' and 'hypolimnion'. Even though the hypolimnion exhibits considerable internal chemical gradients, previous work has shown that the MOB assemblage is fairly homogeneous throughout the hypolimnion (Mayr et al., 2020b). In January, the lake was completely mixed. To be consistent with the previous sampling campaigns we still took two samples from different depths and refer to them as epilimnion and hypolimnion for convenience.

From October 2017 to January 2018 the epilimnion depth in Lake Rotsee gradually increased from 5.5 to 13.7 m (Fig. 1a-d). This process of vertical mixing continuously transferred methane that was stored below the thermocline into the epilimnion above. The gradual progression of the autumn overturn stimulates the growth of a distinct MOB assemblage in the epilimnion above the thermocline in response to an influx of methane from the hypolimnion as shown in previous work of Lake Rotsee (Mayr et al., 2020b; Zimmermann et al., 2019). Despite this continuous supply, measured methane concentrations above the thermocline remained below 1 μM (Fig. 1a-d, orange arrows). The low methane concentrations are an indication of intense methane oxidation by the growing MOB assemblage in the epilimnion. The oxygen concentration shifted from 15% oversaturation in October to 67% undersaturation in December (Fig. 1a-d). Aerobic methane oxidation likely contributed to the oxygen depletion in the epilimnion, which we substantiate with the following calculation: The stoichiometry of microbial methane oxidation is:

$$CH_4 + (2 - y)O_2 \rightarrow (1 - y)CO_2 + yCH_2O^{BM} + (2 - y)H_2O$$

where y is the carbon use efficiency and $CH_2O^{BM}$ designates MOB biomass. Based on theoretical considerations and experimental data, a carbon use efficiency of 0.4 has been reported (Leak and Dalton, 1986). This means that per mole of methane 1.6 moles of oxygen are used. The mixed layer depths for the four sampling campaigns are

269 roughly 6, 10, 12 and 14 m, corresponding to mixed layer volumes of 2.5, 3.7, 4.1 and 4.3 GL in Lake Rotsee.
Multiplying the measured methane oxidation rates in the epilimnion with these volumes results in a total methane
oxidation of 600, 11560, 11800 and 200 mol $d^{-1}$. Integrated over the time period of the four campaigns, this
results in a total of 0.66 Mmol of methane that were oxidized with 1.1 Mmol of oxygen from the epilimnion. In
an average volume of the mixed layer of 3.7 GL with an initial concentration of 340 µM (10.9 mg $L^{-1}$) of oxygen,
this would reduce the oxygen concentration by 180 µM to 160 µM or to about 5 mg $L^{-1}$. Note that possible
oxygen production and exchange with the atmosphere, as well as additional oxygen sinks are not included in
these considerations. In the hypolimnion oxygen concentrations were below the detection limit (20 nM) (Kirf et
al., 2014) from October to December. However, oxygen may be produced in the hypolimnion by phytoplankton
(Brand et al., 2016; Oswald et al., 2015).

The two water bodies above and below the thermocline have distinct biogeochemical conditions posing very
different demands on the ecophysiology of the MOB assemblage. The hypolimnion contained up to several
hundred micromolar of methane but the flux of oxygen into the hypolimnion was limited due to stratification and
low light levels for photosynthesis. On the other hand, the epilimnion contained comparably high oxygen
concentrations, but methane concentrations remained low as methane was supplied slowly and was rapidly
diluted in the large volume of the epilimnion. In addition, the temperature of the epilimnion dropped from 16 °C
to 5 °C, whereas the hypolimnion remained cold (5 - 8 °C). A previous study investigating 16S rRNA genes and
*pmoA* transcripts indeed revealed niche differentiation of the MOB assemblage above and below the oxycline of
Lake Rotsee with a shift in the MOB assemblage during the overturn (Mayr et al., 2020b).

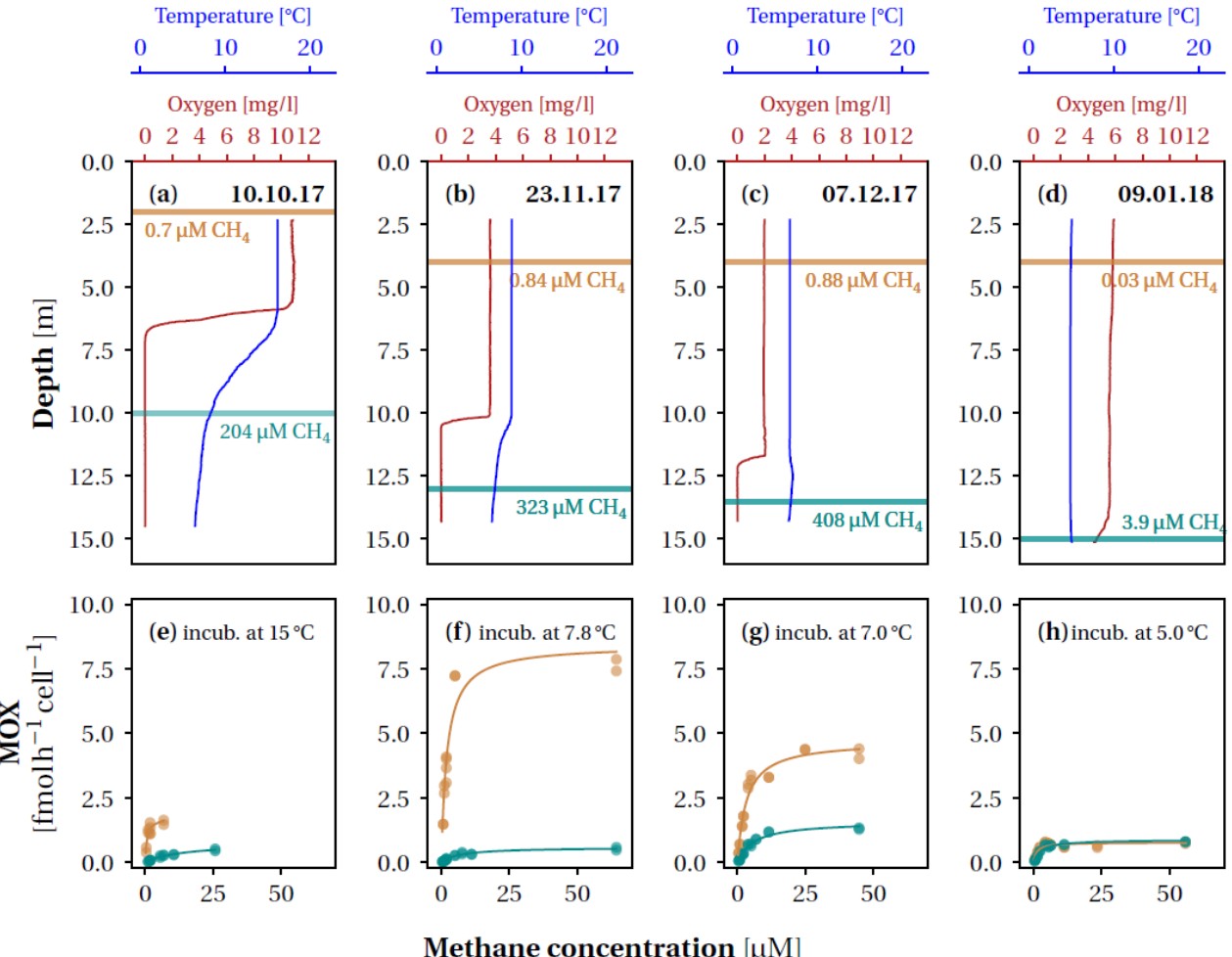

**Figure 1. Substrate concentrations and apparent methane oxidation kinetics during lake overturn in Lake Rotsee. (a - d)**
**Oxygen concentration and temperature profiles during the four field campaigns at the dates indicated. The sampling depths**
**above (orange) and below (cyan) the oxycline are indicated by a horizontal bar. Numbers next to the bars represent methane**
**concentrations in µM at the respective depths. (e - h) Cell-specific methane oxidation rates (MOX) of water samples incubated**
**with different methane concentrations. Lines indicate least-square fits of the Monod kinetics. For each campaign, we**
**incubated samples from both depths close to *in-situ* temperature, given next to the annotation (incub. = incubated).**

## 3.2 Succession of kinetically different microbial communities

Along with the differences in the physical and chemical properties of the two water bodies, we observed a significant difference in the apparent methane oxidation kinetics of the MOB assemblages. From the methane oxidation rates shown in Fig. 1e-h we derived the parameters of Monod kinetics (Fig. 2). In an attempt to measure methane kinetics under as standardized conditions as possible we measured samples from both depths under equal temperature and oxygen conditions. This allowed us to compare the physiological traits of the MOB assemblages above and below the oxycline and to relate these results to the biogeochemical conditions. However, the parameters therefore do not necessarily represent the effective in-situ kinetics.

The curves describing the apparent methane oxidation kinetics of the MOB assemblages above and below the oxycline did not intersect (except at the origin) in October and November (Fig. 1e-g). This means that the MOB assemblage in the epilimnion showed both a higher *affinity* for methane (Fig. 2a) and a higher cell-specific maximum methane oxidation rate (Fig. 2b) than the assemblage below the oxycline. The fact that both *affinity* and maximum rate are higher would by itself suggest that the assemblage in the epilimnion has a competitive advantage over the assemblage in the hypolimnion. This implies that there were likely additional mechanisms or traits, like adaptation to oxygen concentration or temperature (Hernandez et al., 2015; Trotsenko and Khmelenina, 2005), that prevented the epilimnetic MOB assemblage from invading the assemblage in the hypolimnion. We already have strong indications from our previous work that these factors are indeed important (Mayr et al., 2020b, 2020a).

The methane *affinity* of the assemblage in the epilimnion was higher than the methane *affinity* of the assemblage in the hypolimnion, which is in line with the methane-deficient conditions in the epilimnion. Previously, starvation of methane has been shown to decrease the $K_m$ in *Methylocystis* (Dunfield and Conrad, 2000), but in contrast to this study we did not observe a constant specific affinity between epi- and hypolimnion, suggesting that indeed adaptation rather than a starvation response was responsible for the observed low $K_m$ in the epilimnion. The pronounced difference in $K_m$ of the two assemblages in October, when the lake was still stratified, gradually converged during lake overturn from November to January (Fig. 2a). From October to January, the half-saturation constant for methane decreased from 15 to 2.7 µM for the hypolimnetic assemblage, but increased from 0.7 to 1.2 µM in the epilimnion, with higher $K_m$ values in November and December (Fig. 2a). A table summarizing the measured apparent methane oxidation kinetics can be found in Supplementary Tab. 1. The half saturation constants ($K_m$) in the hypolimnion from October to December ($15.2 \pm 7.1$ µM, $7.1 \pm 2.3$ µM, $6.1 \pm 1.7$ µM) were comparable to $K_m$ values of hypolimnion samples (one meter above sediment) in two shallow arctic lakes by Lofton et al. (2014). These authors measured values of $4.45 \pm 2.36$ µM and $10.61 \pm 2.03$ µM. Also in the same range, $K_m$ values of 5.5 µM and 44 µM were measured in the last meter above the sediment in a boreal lake (Liikanen et al., 2002) and similar values were found for lake sediments (Kuivila et al., 1988; Remsen et al., 1989). In contrast, the epilimnion $K_m$ in Rotsee in October was $0.7 \pm 0.5$ µM, which is far lower than $K_m$ values measured in previous studies on lacustrine systems, suggesting a well-adapted MOB assemblage with relatively high *affinity* in the epilimnion. In soils even higher affinities have been measured (0.056 – 0.186 µM) (Dunfield et al., 1999) and a high-affinity *Methylocystis* strain has been found to have a $K_m$ of 0.11 µM (Baani and Liesack, 2008). Even when the lake overturn was ongoing in November and December, $K_m$ values in the epilimnion stayed in the lower range of previously reported $K_m$ values ($2.1 \pm 0.9$ µM, $3.3 \pm 0.9$ µM), which underlines the adaptation of the MOB assemblage to the continuously lower methane concentrations in the epilimnion.

We thus concluded that MOB assemblages displayed a specific adaptation to the prevailing methane concentrations based on the fact that we observed a higher *affinity* (low $K_m$) in the low-methane epilimnion compared to the methane rich hypolimnion as long as stratification is present. That the $K_m$ values of the assemblage in the epilimnion do not match the *in-situ* methane concentrations is not unexpected: In the mixed layer and under the assumption of a steady-state, the flux of methane from the hypolimnion is balanced by the methane oxidation rate. Under these conditions, the *in-situ* methane concentration depends on the half-saturation constant ($K_m$) but should be lower than it (Supplementary Calculation 1). Per definition, the half-saturation constant is the substrate concentration where the growth rate is half the maximum growth rate. Even if the growth rate is only half the maximum growth rate, microbial methane oxidation continues, and methane concentrations decrease to values below $K_m$.

In contrast to the substrate *affinity*, the maximum cell-specific methane oxidation rate $V_{max}$ started at similar levels in the stratified lake (Fig. 2b). As methane entered the epilimnion in November, the cell-specific $V_{max}$ of the MOB assemblage in this layer was almost 15 times faster than the hypolimnion assemblage, which ensured a fast methane oxidation rate in the epilimnion during this critical phase, which has a large potential for methane outgassing to the atmosphere and thus with climate relevance. As a consequence of the high methane oxidation rate, methane concentrations and emissions remain low (Zimmermann et al., 2019). Towards the end of the lake overturn, when the thermocline had moved to 15 m depth and the two MOB assemblages were most likely homogenized, methane oxidation rates decreased again. By contrast, the cell-specific methane oxidation rate in the hypolimnion remained rather constant throughout the overturn from November to December.

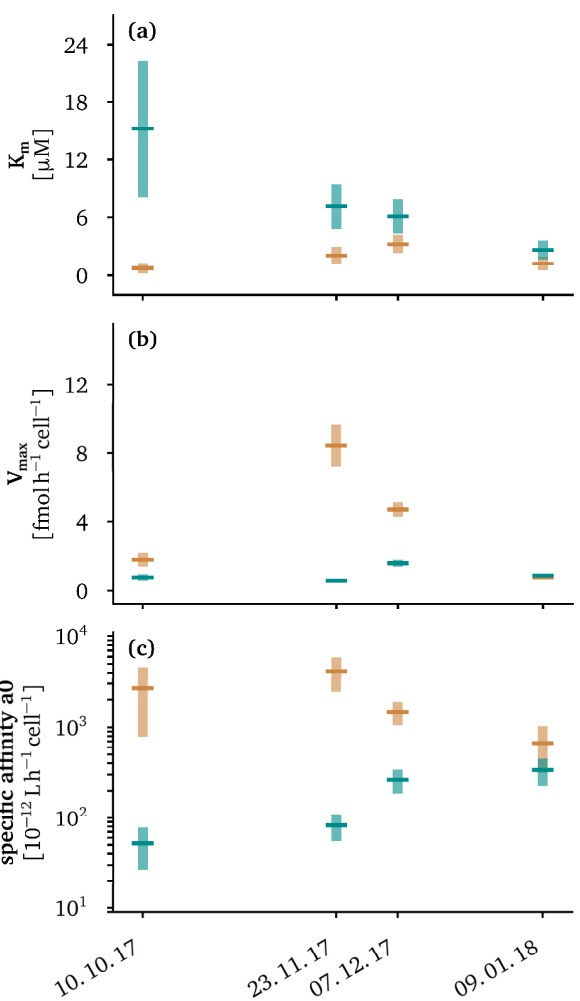

**Figure 2. Apparent kinetic properties of the methanotroph assemblage above (orange) and below (cyan) the oxycline for the four sampling campaigns at *in-situ* temperatures of the oxycline. We plot 95% confidence intervals as light cyan and light orange vertical bars. Average values are plotted as dark green and dark orange lines. The methane oxidation half-saturation constants ($K_m$) are displayed in panel (a), maximum cell-specific methane oxidation rates in panel (b) and specific affinities, defined as the ratio $V_{max}/K_m$, in panel (c).**

The *specific affinity* ($V_{max}/K_m$) is the initial slope of the hyperbolic Monod kinetics (Button et al., 2004) and is a pseudo first order rate constant for the methane oxidation rate at limiting methane concentrations. The *specific methane affinity* of the two communities again started out very differently and gradually converged to very similar kinetic properties (Fig. 2c). The convergence of the *specific affinity* in the epilimnion and in the hypolimnion was driven by changes of both, $K_m$ and $V_{max}$ of the respective MOB assemblages. The final convergence of the *specific affinity* of both assemblages is in good agreement with the fact that the two water masses become increasingly similar in terms of substrate availability and temperature towards the end of the lake overturn. The emerging kinetic properties might therefore be the result of a converging succession of the two MOB assemblages. The *specific affinity* measured for various methanotrophic bacteria are typically in the range of 1 to 40 x $10^{-12}$ L h$^{-1}$ cell$^{-1}$ (Dunfield and Conrad, 2000; Knief and Dunfield, 2005; Tveit et al., 2019) with a few examples where specific affinities of up to 800 x $10^{-12}$ L h$^{-1}$ cell$^{-1}$ were reported (Calhoun and King, 1997; Kolb et al., 2005). The specific affinities of 52 – 338 x $10^{-12}$ L h$^{-1}$ cell$^{-1}$, of the MOB assemblage in the hypolimnion were thus well in the range of these reported values. However, the MOB assemblage in the epilimnion showed

much higher specific affinities suggesting that these assemblages were well adapted to the very methane limited conditions in the epilimnion.

Methanotroph cell counts suggest that both the MOB assemblage above and below the oxycline were actively growing over the course of the overturn (Supplementary Fig. 1). In the epilimnion the abundance of MOB increased from $0.1\times10^5$ to $2\times10^5$ cells mL$^{-1}$ from October to December, below the oxycline the abundance increased from $0.8\times10^5$ to $1.2\times10^5$ cells mL$^{-1}$. The methane oxidation rates of the MOB assemblage in the epilimnion were all below 50 % of $V_{max}$ from October to December. For the MOB assemblage in the hypolimnion, the methane oxidation rates were all above 67 % of $V_{max}$. Even though we don't have enough data points to recognize specific trends, the clear differences in the percentage range suggests that the growth of the MOB assemblage in the epilimnion was generally methane limited during lake overturn, despite their higher methane *affinity*.

**3.3 Dynamics of the MOB assemblage and variants of pMMO**

Methane oxidation during lake overturn was performed by diverse assemblages of MOB that changed considerably over time, as determined by metatranscriptomic analysis (Fig. 3a-c). Thus, the reported apparent kinetics reflect composite properties of the respective assemblage. In line with previous lake studies (Biderre-Petit et al., 2011; Mayr et al., 2020a; Sundh et al., 2005), the majority of *pmoCAB* variants were associated with type Ia MOB (Fig. 3a, green, *Gammaproteobacteria*). Most of these could not be classified at lower taxonomic levels, but group close to reference sequences from various different genera e.g. *Crenothrix*, *Methylobacter*, *Methylovulum* and Aquatic cluster 5 group (Supplementary Fig. 2). However, one sequence variant could be classified as Lake cluster I which was abundant especially in Nov and Dec (Fig. 3 a, b). In addition, one variant associated with *Methyloparacoccus* (type Ib, *Gammaproteobacteria*) and up to two variants affiliated with type II MOB (*Alphaproteobacteria*) were found (Fig. 3a), the latter showed a low abundance and decreasing trend over time. Evidence for the presence or expression of previously described high affinity pMMO (Baani and Liesack, 2008) was not found in the metagenomic or metatranscriptomic dataset. We detected sMMO genes (*mmoXYZ*) but transcription was very low (maximum of 6 TPM per sample, Supplementary Tab. 2) compared to pMMO. This raises the question under which conditions MOB express sMMO. On the transcript and peptide level the expression of this enzyme is often very low or undetectable under environmental conditions (Cheema et al., 2015; Dumont et al., 2013; Taubert et al., 2019). Our results suggest that conventional pMMO was the main enzyme responsible for methane oxidation under different methane concentrations and environmental conditions in the lake water column.

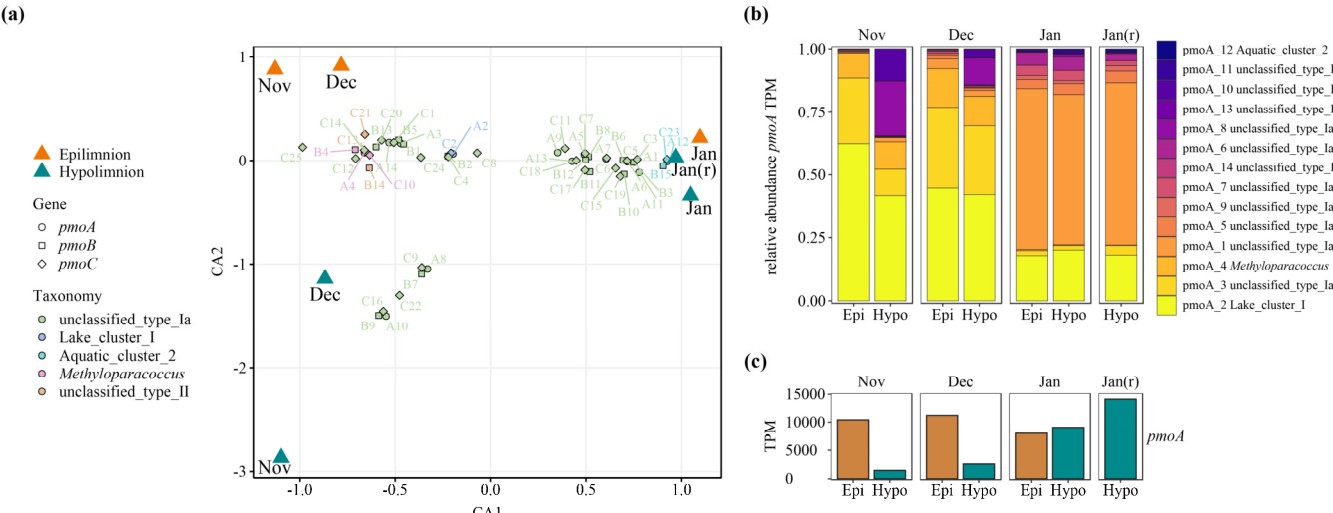

**Figure 3. Transcribed gene variants encoding pMMO in November, December and January 2017/2018 during overturn in Rotsee. The January hypolimnion sample was extracted from two filters and the replicate is labelled as Jan(r).** *pmoCAB* **variants were assembled from metagenomic and metatranscriptomic samples and mapped at 99% identity. The samples originate from the same depths and dates as shown in Fig. 1. From October no metatranscriptomes are available. (a) Correspondence analysis (CA) of combined, square root transformed** *pmoCAB* **TPM (transcripts per million) data. Sample scores are shown as big coloured triangles (orange = epilimnion and cyan = hypolimnion). Sequence variant scores are shown as smaller symbols; circle =** *pmoA***, square =** *pmoB***, diamond =** *pmoC***. Identified variants of each gene are numbered and abbreviated as** *pmoA* **= A,** *pmoB* **= B,** *pmoC* **=**

C. Colours depict the taxonomic classification of the sequence variants based on a phylogenetic tree (Supplementary Fig. 2). (b) Relative abundance of gene variants of *pmoA* based on transcripts per million (TPM). The corresponding figures for *pmoB* and *pmoC* variants are shown in Supplementary Fig. 3. (c) *pmoA* shown as summed TPM of all variants for epi- and hypolimnion (orange and cyan, respectively). Epi = epilimnion, Hypo = hypolimnion. The figures for *pmoB* and *pmoC* are shown in Supplementary Fig. 3.

In November, and to a lesser degree in December, the composition of transcribed *pmoCAB* gene variants differed between epi- and hypolimnion, with some variants (e.g. pmoA_8 and 10, pmoB_7 and 9, pmoC_9, 16, 22) being confined to the hypolimnion (Fig. 3a, b). In November and December the relative transcript abundance of *pmoCAB* was higher in the epilimnion, but the activity in the hypolimnion increased over time and was similar in both depths in January (Fig. 3c, Supplementary Fig. 3c,d). The difference in gene transcription reflects changes in the transcriptionally active MOB assemblage, which may explain the observed differences in the apparent methane *affinity* (Fig. 3a). Notably however, a prominent proportion of the *pmoCAB* gene variants transcribed in the epilimnion were also present in high relative abundance in the hypolimnion, which may reflect an increasing influence of the highly transcriptionally active epilimnion assemblage (Fig. 3b, c, Supplementary Fig. 3) on the hypolimnion assemblage during lake overturn. Thus, a contribution of organisms present in different physiological states (e.g. starvation) to the difference in apparent kinetics cannot be ruled out by this data. Unfortunately, we lack information on the assemblage for the October sampling where the half saturation constants differed most between epi- and hypolimnion. However, based on observations of the overturn period the year before (Mayr et al., 2020b), it can be assumed that the two layers harboured distinct MOB assemblages also in October, likely with less species overlap.

Similar to observations made on MOB communities the year before (Mayr et al., 2020b), the *pmoCAB* transcript variants confined to the hypolimnion (e.g. pmoA_8) decreased over time and did not establish in the epilimnion (Fig. 3b, Supplementary Fig. 3c,d). In January the *pmoCAB* composition became almost indistinguishable between epilimnion and hypolimnion (Fig. 3a,b). At the same time their apparent kinetic properties became increasingly similar as well (Fig. 2), which is also in line with the advanced stage of the mixing processes (Fig. 1d). The replicate sample Jan(r) showed a very similar composition providing confidence in the metatranscriptomic analysis, but also showed some variability concerning the summed TPM abundance of the *pmoCAB* variants (Fig. 3c). From December to January a strong shift in the MOB assemblage towards dominance of pmoA_1, pmoB_3 and pmoC_3 occurred (Fig. 3a, b). The shift of the MOB assemblage was accompanied by a drop in temperature and rise in oxygen, which are probable drivers of MOB succession in addition to methane availability (Hernandez et al., 2015; Oshkin et al., 2015; Trotsenko and Khmelenina, 2005).This did however not lead to much change in the methane *affinity* (Fig. 2), suggesting that different MOB assemblages can have similar methane affinities. Nevertheless, we hypothesize that the composition of *pmoCAB* rather than the summed TPM may be important for explaining the kinetic properties. With this shift, we also observed a decrease in $V_{max}$ per cell (Fig. 2b). In agreement with observations made the year before (Mayr et al., 2020b), we attribute the decrease in $V_{max}$ per cell to a shift from growth-oriented MOB dominating the bloom phase to a late-successional MOB assemblage adapted to cold temperatures. Overall, the metatranscriptomic analysis supports the hypothesis that the observed differences in apparent methane oxidation kinetic parameters between water layers and over time have a basis in compositional differences of the transcriptionally active MOB assemblages.

## 4 Conclusions

In Lake Rotsee, as in many other stratified lakes (Bastviken et al., 2004; Borrel et al., 2011), the high methane availability in the hypolimnion contrasts with low methane availability in the epilimnion. Therefore, we hypothesized that the resident MOB assemblages are adapted to the local conditions. Our field study revealed a high level of adaptation of the MOB assemblage: the $K_m$ was 20 times higher in the hypolimnion than in the epilimnion during stable stratification. Transcribed methane oxidation genes differed as well, indicating that methane *affinity* is one important trait structuring MOB assemblages in this system. The MOB assemblage and its apparent kinetic parameters adapted rapidly to changing conditions in the epilimnion. In October, the low epilimnion $K_m$ suggested an adaptation to low methane concentrations. During the autumn overturn, *affinity* decreased slightly but remained above hypolimnion values, reflecting persistently low methane concentrations that suggest methane-limited growth despite higher methane input. We observed increased $V_{max}$ in the epilimnion

during November and December ensuring a fast methane oxidation rate in this period with continuous transport of methane into the epilimnion. By contrast, in the hypolimnion methane concentrations during overturn exceeded the $K_m$ several-fold suggesting that MOB growth was not limited by methane concentrations.

Our transcriptomic analysis revealed that the variations in methane *affinity* were linked to transcribed, and thus likely expressed, *pmoCAB* variants. We also found that pMMO appeared to be the dominant methane monooxygenase throughout and found no evidence for shifts between sMMO and pMMO transcription as hypothesized previously (Semrau et al., 2018). Nor could we observe any of the previously described high-*affinity* pMMO variants, which suggests considerable, so far unappreciated variability in apparent pMMO kinetics. Further research will be needed to obtain kinetic data on individual pMMO variants and to better understand the physiological basis of the apparent methane oxidation kinetics. The provided apparent kinetic parameters for lake MOB assemblages will inform future trait or process-based models of the MOB assemblage and methane emissions. In summary, our work demonstrates that differential methane availability governed by lake mixing regimes created niches for MOB assemblages with well-adapted methane-oxidation kinetics in Lake Rotsee, a mechanism that possibly applies to many seasonally stratified lakes in which vertical structure and temporal succession of MOB may be similar.

## Figure captions

**Figure 1. Substrate concentrations and apparent methane oxidation kinetics during lake overturn in Lake Rotsee. (a - d) Oxygen concentration and temperature profiles during the four field campaigns at the dates indicated. The sampling depths above (orange) and below (cyan) the oxycline are indicated by a horizontal bar. Numbers next to the bars represent methane concentrations in μM at the respective depths. (e - h) Cell-specific methane oxidation rates (MOX) of water samples incubated with different methane concentrations. Lines indicate least-square fits of the Monod kinetics. For each campaign, we incubated samples from both depths close to *in-situ* temperature, given next to the annotation (incub. = incubated).**

**Figure 2. Apparent kinetic properties of the methanotroph assemblage above (orange) and below (cyan) the oxycline for the four sampling campaigns at *in-situ* temperatures of the oxycline. We plot 95% confidence intervals as light cyan and light orange vertical bars. Average values are plotted as dark green and dark orange lines. The methane oxidation half-saturation constants ($K_m$) are displayed in panel (a), maximum cell-specific methane oxidation rates in panel (b) and specific affinities, defined as the ratio $V_{max}/K_m$, in panel (c).**

**Figure 3. Transcribed gene variants encoding pMMO in November, December and January 2017/2018 during overturn in Rotsee. The January hypolimnion sample was extracted from two filters and the replicate is labelled as Jan(r). *pmoCAB* variants were assembled from metagenomic and metatranscriptomic samples and mapped at 99% identity. The samples originate from the same depths and dates as shown in Fig. 1. From October no metatranscriptomes are available. (a) Correspondence analysis (CA) of combined, square root transformed *pmoCAB* TPM (transcripts per million) data. Sample scores are shown as big coloured triangles (orange = epilimnion and cyan = hypolimnion). Sequence variant scores are shown as smaller symbols; circle = *pmoA*, square = *pmoB*, diamond = *pmoC*. Identified variants of each gene are numbered and abbreviated as *pmoA* = A, *pmoB* = B, *pmoC* = C. Colors depict the taxonomic classification of the sequence variants based on a phylogenetic tree (Supplementary Fig. 2). (b) Relative abundance of gene variants of *pmoA* based on transcripts per million (TPM). The corresponding figures for *pmoB* and *pmoC* variants are shown in Supplementary Fig. 3. (c) *pmoA* shown as summed TPM of all variants for epi- and hypolimnion (orange and cyan, respectively). Epi = epilimnion, Hypo = hypolimnion. The figures for *pmoB* and *pmoC* are shown in Supplementary Fig. 3.**

## Data availability

Raw reads of the sequencing project were submitted to the European Nucleotide Archive under project number PRJEB35558. Methane concentrations, scintillation counts, methane oxidation rates, estimated kinetic parameters and the identified nucleotide sequences encoding MMO are available at the EAWAG repository under https://doi.org/10.25678/0001fa (Mayr et al., 2019).

## Author contribution

MJM and MZ contributed equally to this work. MJM, MZ, and HB conceptualized the study and MJM, MZ and JD carried out the investigation. MJM and MZ curated, analysed and visualized the data. MJM and MZ wrote the original draft of the manuscript with contributions from BW, HB and JD. Funding was acquired by HB.

## Competing interests

The authors declare that they have no conflict of interest.

## Acknowledgements

This research was funded by the Swiss National Science Foundation (grant CR23I3_156759), by ETH Zurich and Eawag. We are grateful to Andreas Brand for his support and advice in the early stages of the project, to Carsten Schubert, Serge Robert and Daniel Steiner for the possibility and the support to use the equipment for radioisotope and methane measurement. We are also grateful to Lea Steinle for sharing her expertise on how to handle the radiolabelled methane. We would like to thank Karin Beck and Patrick Kathriner for technical assistance during field work and laboratory analysis. We thank Feng Ju and Robert Niederdorfer for advice on the

496 bioinformatics analysis. Sequencing data were analysed in collaboration with the Genetic Diversity Centre
(GDC) of ETH Zurich.

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

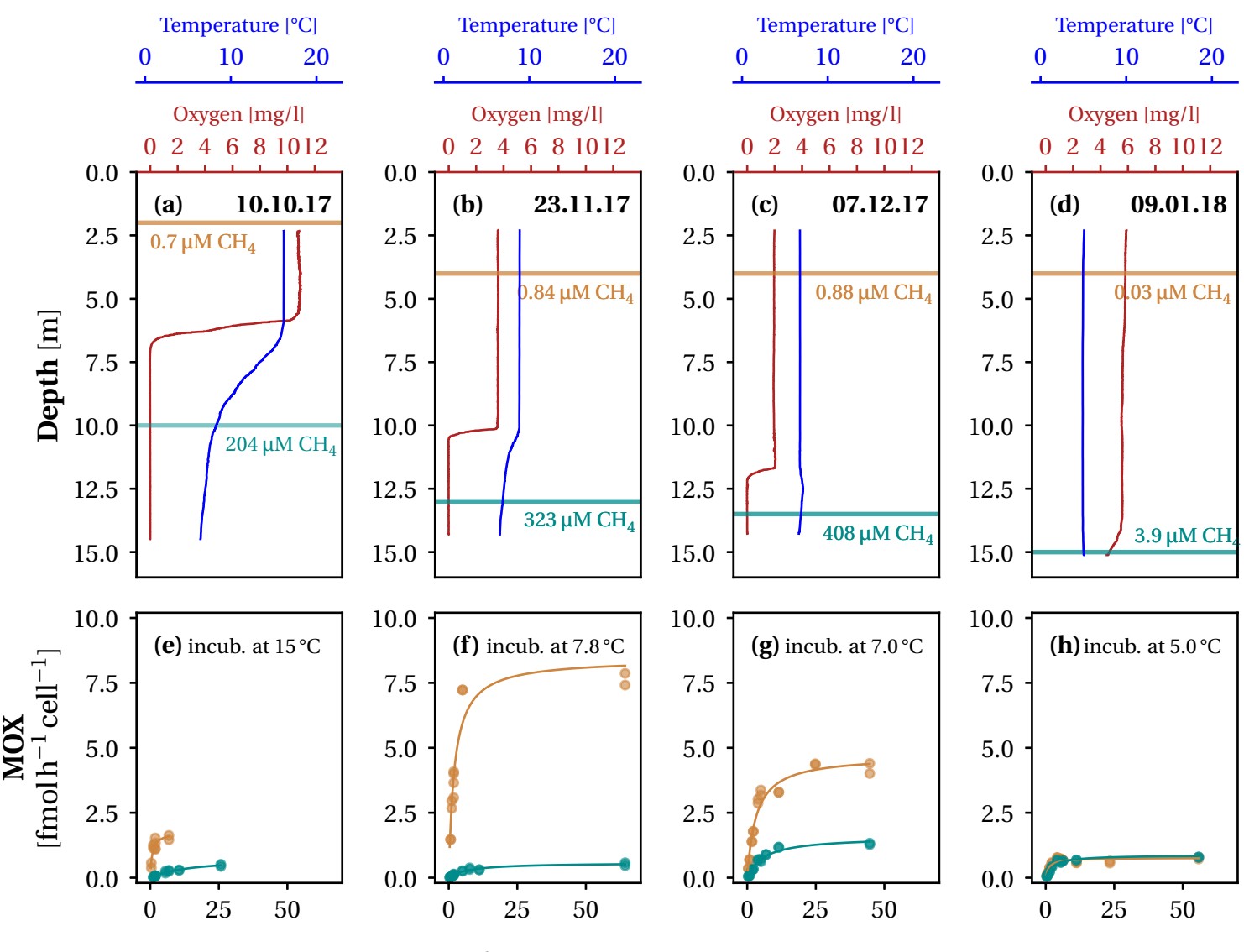

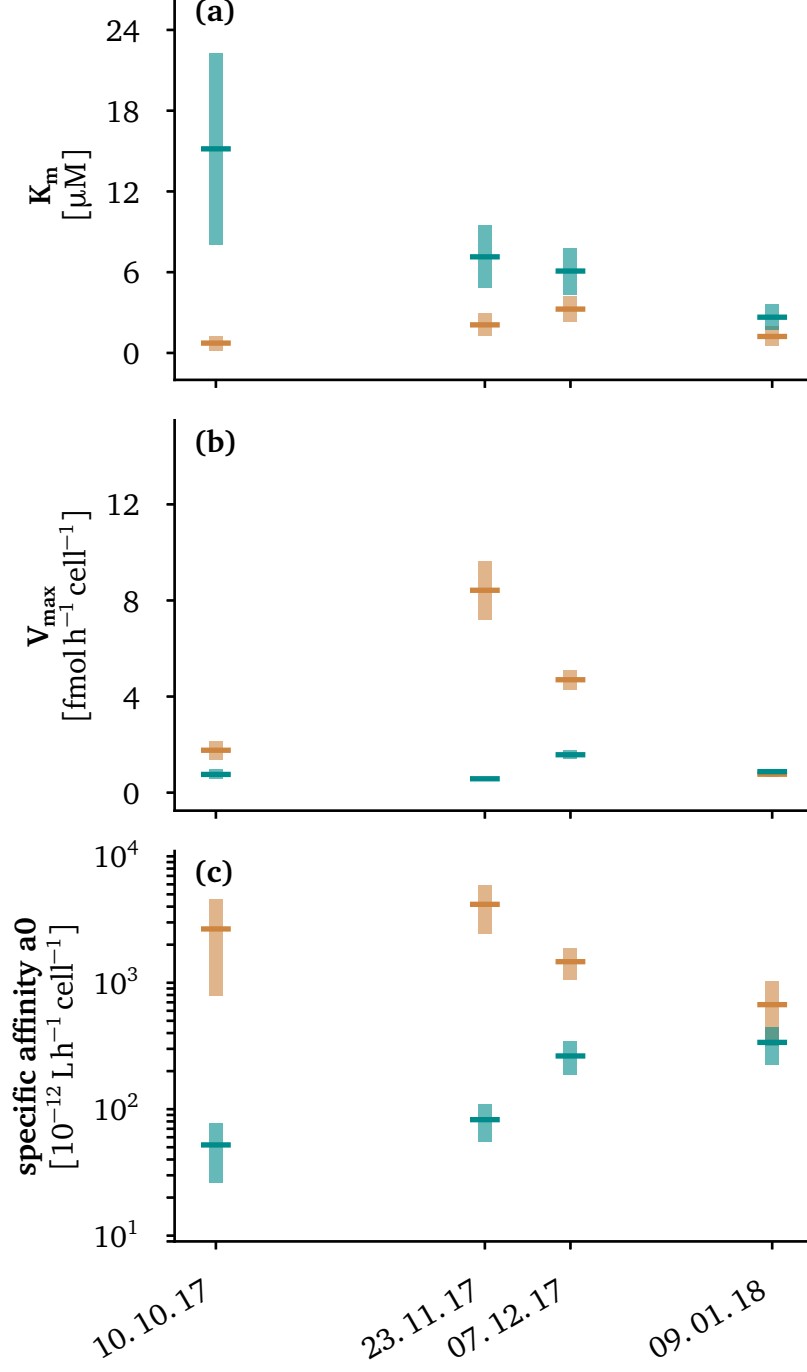

**(a)**

**(b)**

**(c)**