# Peer review of "Lake mixing regime selects apparent methane-oxidation kinetics of the methanotroph assemblage"

_Biogeosciences, 2019_

## Referee Comment (RC1) · Anonymous Referee #1 · 10 Feb 2020

The manuscript on "Lake mixing regime selects methane oxidation kinetics of the methanotroph assemblage" by Mayr and Zimmermann et al, is a well written manuscript on the ecophysiology of methanotrophic bacteria.

I only have the following remarks: Method section on the kinetics: - As you describe in detail how you eliminated outliers from the calculations I was wondering what the percentage of outliers was? From my (more marine) experience it is difficult to get good kinetic data, as many of my "Kinetic incubations" only gave erratic results.... -Filtering the samples water on the 0.2 ym filters. How long did it take to filter 1 - 2 liters on such a filter?? To my experience this may last very long.... Thus, did
you use any prefilters? And could the RNA composition change of this presumably longer time?? Result /Discussion - Figure 1: I think it is essential also to show the methane concentrations, in situ MOXrates and also the cell numbers in one figure, as the "environmental background information". The second part of fig. 1 the kinetics should go in a separate figure, as this is more an experimental aspect. - Comment on figure 2: to me another way of seeing the data is, that in the epilimnion Km is much higher in October, but is getting more and more similar to the hypolimnion. Thus it is not a mixing of two compartments but more of an approximation of the epilimnetic traits to the hypolimnetic ones ??

---

## Referee Comment (RC2) · Anonymous Referee #2 · 12 Feb 2020

Mayr, Zimmermann and colleagues studied the methane oxidation kinetics in the epi- and hypolimnion of a eutrophic lake during autumn/winter lake overturn and report changing methane uptake kinetics. Likewise, changes in pmoC, A and B gene expression profiles were observed, indicating adjustments in the active methanotrophic community in dependence on methane availability. I see value in the presented work, but also limitations and open questions that would have to be clarified.

First of all, I request the authors to point out that they measured apparent methane oxidation kinetics. This should be clearly indicated throughout the manuscript.

In this study, all conclusions are derived from 1 - 2 l of water per sample, taken at

four different time points of the epilimnion and hypolimnion, respectively. However, no replicate samples were taken per layer. I find this hardly acceptable. How representative are the findings of 1 l water for the whole stratification layer of a lake? Can the authors be sure that the differences they see are indeed related to the respective water bodies? Already the molecular analysis of a second filter, which is a methodological replicate that was included for one sample, shows some differences (Fig. 3). Thus, I find it largely impossible to relate differences in the active methanotrophic community to stratification, especially in December and January and especially for pmoB and C (Fig. 3; statements l. 266-267), without knowing anything about the biological variation within a layer. The repeated measurements over time provide some evidence, but do not solve this issue when it comes to minor differences between specific samples. Besides, the reason for including one experimental replicate (January, hypolimnion) or the conclusions derived from this sample are not mentioned anywhere.

How do the authors know that they had a representative sample from the hypolimnion at the last sampling date? There is no change in temperature evident and the decline in oxygen concentrations does not reach oxygen concentrations as low as at the earlier time points. Likewise, methane concentration in this sample is not as high as in the other samples from the hypolimnion. Thus, it appears that the sample was not taken at appropriate depth to be comparable with the others.

The conclusion about the specific enrichment of well-adapted methanotrophs with particular methane oxidation kinetics (l. 23) is conceivable, but should be drawn more carefully, because it remains unclear whether the observed kinetics are indeed adaptations of particular competitive methanotrophs under oligotrophic conditions, especially with regard to affinity. As only apparent parameters could be estimated, it remains unclear whether the methane monooxygenase of the respective organisms has indeed a higher affinity (lower Km) and is thus more competitive. It should be kept in mind in this context that a low apparent Km is not necessarily a specific adaptation to low methane concentrations, but can be the result of starvation (see Dunfield and Conrad

2000, AEM).

To determine methane uptake kinetics (Fig. 2), the samples were apparently incubated at the temperatures measured in the epilimnion. However, samples from the hypolimnion encounter much lower temperatures in autumn. How does that affect comparability of the obtained results and conclusions about in situ conditions? This should be taken into account.

Related to this point: Considering that altered temperature and oxygen conditions were used to characterize the methane uptake kinetics in vitro, to what extent can the findings be translated to in situ conditions, considering that these factors can affect the measured Km and Vmax (see the study of Thottathil et al 2019, who report that increasing oxygen concentrations in lake water can reduce maximum methane oxidation rates; doi.org/10.1007/s10533-019-00552-x). Is it conceivable that Vmax in the hypolimnion is underestimated when determining oxidation rates at higher oxygen concentrations in vitro?

I find it very unfortunate that the identification of methanotrophs stops at the level "type Ia, type Ib, type II". The sequence information should provide more detailed information about the identity of the methanotrophs. At least for pmoA comprehensive datasets are available covering besides cultivated strains diverse groups of uncultivated taxa, so that more information could have been extracted here to identify conspicuous taxa.

Specific comments: l. 19 and 291: According to the data in table S1, the difference in Km is 20-fold, not 2 orders of magnitude

l. 25: Where in the presented work is it shown or discussed that 90% of the methane are removed? It appears that this is not a conclusion that is derived from the presented work.

l. 65: Metagenomic data were used as a basis for the metatranscriptomic data analysis, but are not presented independently; thus, I would not emphasize the metagenomics

approach here for the analysis of MOB assemblages.

l. 73: Five campaigns in autumn 2017 does not appear correct (three samplings in 2017 and one in 2018 according to the presented results)

l. 74-75: More measured parameters are given here than presented; harmonize.

l. 78: I do not find any helpful information about the radio isotope tracer technique in Steinle et al 2015. While the cited references enabled me to understand how methane oxidation rates were determined, they do not allow me to evaluate whether/how this procedure can be used to survey methane oxidation kinetics.

l. 80: How much methane was in this mixture?

l. 100-106: The authors describe different criteria that were used to identify and eliminate outliers here. Point four states that data points were removed in case less then two replicates remained. According to l. 93, duplicates were prepared. Does that mean that data for a specific methane concentration were lost each time one of the two replicates was identified as outlier? In this context, it is also unclear what Fig. 1 e-h shows. Do the presented data points represent individual measurements or are these mean values of the two replicates? Sometimes, I see two data points at a specific concentration, but sometimes I see only one point. Please clarify.

It would be valuable to know how many high-quality reads the authors generated per sample in the metagenomic and metatranscriptomic analysis, respectively.

l. 157: Why three samples in October; to my understanding there should be one from the epilimnion and one from the hypolimnion per point of time.

l. 163: Can a few words be added to describe this custom database? How was it set up? What type of data does it include?

l. 202-205, l. 295 and perhaps elsewhere: wording: do the authors refer to Km or a0 here when talking about affinity?

l. 204-207: I cannot follow argumentation here. And how do the authors explain that the organisms with the higher Vmax and lower Km disappear in January (Fig. 1h), although they should have a competitive advantage?

l. 241: A range of 1 – 40 is a bit outdated. Atmospheric methane oxidizers in soil are meanwhile known to have a0s values with up to 195 x 10-12 L/cell*h (Tveit et al) and in upland soils, estimates are ranging up to 800 x 10-12 L/cell*h (Kolb et al 2005; doi:10.1111/j.1462-2920.2005.00791.x)

l. 248-250: I find the 25% and 93% values critical here, because huge differences are observed at the individual time points. Especially the 93% value appears to be strongly affected by the huge difference observed in December.

l. 254: What do the authors mean with aggregate properties here? What aggregates do they refer to?

l. 256 – 258: It would be very valuable if the described findings could be seen in Figure 3.

l. 292-293: The transcription of genes does not relate to enzyme affinity or apparent Km values; thus, I cannot follow argumentation here.

l. 301: I do not necessarily agree to the term "entirely" in the context with "kinetic traits"; other environmental conditions may have affected the kinetic parameters. Please keep in mind that you can only measure apparent parameters, not enzyme kinetics.

l. 303-304: Please note that Methylocapsa gorgona does not possess a second pmoA gene for "high-affinity oxidation" despite being able to live on very low methane concentrations (Tveit et al).

References: The reference list does not allow to differentiate publications (e.g. Mayr et al 2019a, b, c). The reference list lacks information about the year the work has been published and the indices a,b,c.

Figure 1: The axis showing oxygen concentrations should have a more increments.

Figure 2: explain error bars

Figure 3: The distinction by color is difficult in plots a1-c1; why not choosing more distinct colors / a broader range of colors per plot? This is of particular importance, as the relative abundances cannot be taken from Table S2 without additional calculations. It is currently impossible to identify type Ib or type II methanotrophs based on the color code and without further invest. However, as pointed out above, it would be even more valuable if more taxonomic information could be provided.

Table S3: Provide reference for Knief et al 2015.

---

## Author Comment (AC1) · 3 Mar 2020

Reply to Reviewer 1: The manuscript on "Lake mixing regime selects methane oxidation kinetics of the methanotroph assemblage" by Mayr and Zimmermann et al, is a well written manuscript on the ecophysiology of methanotrophic bacteria.

Answer: Thank you for the positive assessment.

I only have the following remarks: Method section on the kinetics: - As you describe in detail how you eliminated outliers from the calculations I was wondering what the percentage of outliers was? From my (more marine) experience it is difficult to get

good kinetic data, as many of my "Kinetic incubations" only gave erratic results:

Answer: The criteria eliminated 33% of the data. We prepared incubations in triplicates except for the first field campaign where we only prepared duplicates. Except for the first sampling date, we measured each incubation replicate twice. This resulted in 594 measurements, 72 single measurements and 261 measurement duplicates. We averaged "measurement replicates" resulting in 333 data points. From these 333 data points, we removed outliers as described. In total only 221 datapoints were considered in the final analysis (66% of all datapoints without measurement replicates). For the five individual outlier criteria (line 100 – 106), the percentages of detected outliers are:

(1) water fraction radioactivity that was outside $2\sigma$ from the average water fraction radioactivity of all replicates: 4%

(2) water fraction radioactivity that was not above $2\sigma$ from the background water fraction radioactivity: 4%

(3) less than two replicates after the removal of outliers: 3%

(4) resulting methane oxidation rate outside $2\sigma$ from the average methane oxidation rate of all replicates: 19%

(5) methane oxidation rate that was higher than the methane oxidation rate measured for the replicates with the highest methane concentration: 3%

So indeed, the outlier rate is fairly high and this is why we established a clear set of numeric criteria and a large number of measurements to obtain reliable data. The high percentage of outliers for criteria 4 is related to the fact that methane oxidation rates are associated with a higher error than individual measurements because they are computed from multiple individual measurements. The $2\sigma$ approach is one recommended approach for outlier detection (see e.g. Leys et al. 2013) We propose to show the elimination (in %) in the revised manuscript and to further clarify the procedure and its motivation (also with regards to a comment by reviewer 2).

**BGD**

Filtering the samples water on the 0.2 ym filters. How long did it take to filter 1 – 2 liters on such a filter?? To my experience this may last very long: : :. Thus, did you use any prefilters? And could the RNA composition change of this presumably longer time??

Answer: We did not record the exact duration but in our experience filtration takes typically less than 10 and always less than 15 minutes. In order to retrieve enough RNA for metatranscriptomics and at the same time reducing filtration time as much as possible we used large (142 mm) diameter filters. We did not use prefiltration since some filamentous methanotrophs can be quite large (length of up to >100 $\mu$m (Oswald et al. 2017). We took the samples with a niskin bottle and we connected the tubing for the filtration device directly to the niskin bottle. The sample was kept shaded (inside the niskin bottle) and did not experience major temperature changes in the cool autumn/winter weather of the sampling season for this experiment. Thus, changes from the in-situ transcriptional profile are expected to be minor. In-situ filtration might still be preferable, but the equipment for this was not available to us at the time of this work. We will note the approximate time and diameter of the filter to a revised manuscript and other detail to clarify the steps that were taken to insure rapid preservation of the transcriptome.

Result /Discussion - Figure 1: I think it is essential also to show the methane concentrations, in situ MOXrates and also the cell numbers in one figure, as the "environmental background information".

Answer: Methane concentrations are already shown in Fig. 1 (numbers with arrows) we only measured at the sampling depths in this study. The cell numbers that we used in the calculations are shown in Supplementary Table 1, but we will include this data either in the main text or in a figure or table for a revised manuscript.

The second part of fig. 1 the kinetics should go in a separate figure, as this is more an experimental aspect.

Answer: We used this format deliberately to directly connect the core (kinetic) data

of the study with the graphs showing the environmental situation, which we hoped would make it easier for readers to understand how these are linked. We can certainly separate these parts into independent figures in the revision if the reviewer and editor think this would be a better solution.

- Comment on figure 2: to me another way of seeing the data is, that in the epilimnion Km is much higher in October, but is getting more and more similar to the hypolimnion.

Answer: There might be a misunderstanding here? The epilimnion data are shown in orange, and thus are actually lowest in October. (See Fig. legend)

Thus it is not a mixing of two compartments but more of an approximation of the epilimnetic traits to the hypolimnetic ones ??

Answer: Indeed, as we show in work now in press at Communications Biology (Mayr et al. 2020) (https://doi.org/10.1101/707836) there is more going on than simple mixing, i.e. complex dynamics of the community, and species mixed in from the hypolimnion typically do not establish in the mixed layer. This was also reflected in the transcriptomes obtained for this work. By the end of the mixing process however, almost the entire water mass was indeed mixed, and we observed only a small remnant of the hypolimnion. We did briefly outline our understanding of the dynamics (L289-300) but will seek to improve on this in the revision.

References:

Leys, C., C. Ley, O. Klein, P. Bernard, and L. Licata. 2013. Detecting outliers: Do not use standard deviation around the mean, use absolute deviation around the median. Journal of Experimental Social Psychology 49:764–766.

Mayr, M. J., M. Zimmermann, J. Dey, A. Brand, B. Wehrli, and H. Bürgmann. 2020. Growth and rapid succession of methanotrophs effectively limit methane release during lake overturn. Communications Biology in press.

Oswald, K., J. S. Graf, S. Littmann, D. Tienken, A. Brand, B. Wehrli, M. Albertsen, H.

Daims, M. Wagner, M. M. M. Kuypers, C. J. Schubert, and J. Milucka. 2017. Crenothrix are major methane consumers in stratified lakes. ISME Journal 11:2124–2140.

Please also note the supplement to this comment:
https://www.biogeosciences-discuss.net/bg-2019-482/bg-2019-482-AC1-supplement.pdf

---

## Author Comment (AC2) · 4 Mar 2020

**Reply to Reviewer 2:**

Mayr, Zimmermann and colleagues studied the methane oxidation kinetics in the epiand hypolimnion of a eutrophic lake during autumn/winter lake overturn and report changing methane uptake kinetics. Likewise, changes in pmoC, A and B gene expression profiles were observed, indicating adjustments in the active methanotrophic community in dependence on methane availability. I see value in the presented work, but also limitations and open questions that would have to be clarified.

Answer: Thank you for your valuable comments. In the following we try to answer and clarify all open questions and limitations raised.

First of all, I request the authors to point out that they measured apparent methane oxidation kinetics. This should be clearly indicated throughout the manuscript.

Answer: We agree. In a revised manuscript we would change to "apparent methane oxidation kinetics" throughout the text.

In this study, all conclusions are derived from 1 - 2 I of water per sample, taken at C1 four different time points of the epilimnion and hypolimnion, respectively. However, no replicate samples were taken per layer. I find this hardly acceptable. How representative are the findings of 1 I water for the whole stratification layer of a lake? Can the authors be sure that the differences they see are indeed related to the respective water bodies? Already the molecular analysis of a second filter, which is a methodological replicate that was included for one sample, shows some differences (Fig. 3). Thus, I find it largely impossible to relate differences in the active methanotrophic community to stratification, especially in December and January and especially for pmoB and C (Fig. 3; statements I. 266-267), without knowing anything about the biological variation within a layer. The repeated measurements over time provide some evidence, but do not solve this issue when it comes to minor differences between specific samples.

Answer: Horizontal mixing in lakes is strong (Lerman and Chou, 1995, page 86), especially in stratified lakes and horizontal variation within a lake (excluding near-shore water or parts with limited water exchange) is expected to be small (Yannarell and Triplett 2004) in comparison with the vertical variation during stratification and temporal variation of the mixed layer. Taking one profile, usually close to the deepest point is therefore common practice for studies in smaller lakes or lake surveys (Oswald et al. 2017). Sample volume per depth is then typically decided by the requirements of the analytical methods rather than from concerns about the volumes representativeness.

In our earlier study on Rotsee we investigated the general population structure around
the oxycline and later the community composition change over time during lake overturn in considerable detail. We found that especially the mixed surface layer is a very homogeneous environment considering both environmental conditions and the methanotroph assemblage (Mayr et al. 2020a). Therefore, we think that the epilimnion measurements are indeed representative for the mixed layer (epilimnion in this study) and only very small variabilities would be expected from multiple measurements. The focus of this study was not on variability within the mixed layer or within hypolimnion, but rather to demonstrate that differences in kinetic parameters occur at all, in contrasting microbial communities within one lake which has not been demonstrated before. Therefore, although we did not study the variation within a layer in this study, we had considerable previous knowledge on the same lake from the years before and designed the study on this basis.

The hypolimnion during stratification by definition shows very different conditions to the epilimnion, but gradients of methane and other parameters can be observed within the hypolimnion. Therefore, whereas the hypolimnion is less homogeneous than the epilimnion, the difference to the epilimnion is clear based on both, the environmental conditions (orders of magnitude higher methane concentrations) and a different methanotrophic assemblage (this study and Mayr et al. 2020a, 2020b).

Regarding the request for (biological) replication of the kinetic measurements, we have to note that, although measuring different depths and or replicates from the layers would be favourable, we had to adjust the amount of incubations to our handling limit of incubations. We decided to improve the replication for the respective depths to have enough data points to determine the affinity of the respective depth (see our answer on outlier elimination to the other reviewer). Thus, in order to have enough different methane concentrations and technical replication we could only analyse 2 depths. This decision was also based on the results obtained in our previous study of the lake overturn period as discussed above (Mayr et al. 2020a, 2020b). Therefore, we could not have analysed more replicates or depths on one date due to handling
limitation of number of incubations, but we believe that the multiple time points provide confidence in our measurement. The reviewer may note that the existing studies on apparent MO kinetics in lakes were indeed done on lower numbers of samples.

We agree that the difference between epi- and hypolimnion become smaller in December and January, which is expected as the epilimnion is increasing in size and stratification of the lake weakens. In January the hypolimnion is almost gone and only small differences remain between epi- and hypolimnion, both in terms of methane concentration and methanotroph composition. We agree that the characteristics of the epiand hypolimnion change over time (which is expected and was part of our research question) and use the terms epi- and hypolimnion more to clarify where the sample was taken, rather than to imply homogeneity within the category. Because there is a temporal variation, we also do not give average values for the epi- or hypolimnion. We agree that the term hypolimnion for the bottom sample in January might be misleading, because the lake has almost completely mixed, and the strong difference seen between epi- and hypolimnion before are not there anymore.

We will make the different characteristics of epi- and especially hypolimnion clearer in a revised manuscript. We will also tone down and clarify in methods and/or discussion section that the hypolimnion before lake overturn has some internal gradients. Further, in a revised manuscript we will better link and explain our study design to our previous work, in which we did a comprehensive analysis of methanotrophs during lake overturn with multiple measurements of the same layer.

Besides, the reason for including one experimental replicate (January, hypolimnion) or the conclusions derived from this sample are not mentioned anywhere.

Answer: We included this replication in order to increase confidence in our measurements. It is not easy to obtain enough RNA for metatranscriptomics from our lake samples, which we tried to optimize as much as possible considering the trade-off between 1) filtration time and 2) obtaining enough sample for RNA extraction and subsequent
sequencing. Because both effort and costs are high, we included only one replicate, although we agree more would be favourable. Typically, RNA based measurements show a somewhat larger variability than DNA based sequencing results, therefore we think that our replicate shows that the results are very reproducible for a metatranscriptomics sample, but that small differences between samples should not be overemphasized. That the three January samples are basically indistinguishable based on metatranscriptomics is not surprising as the mixing process is almost complete in January and the epi- and hypolimnion depths are similar at this date (see answer above). We will add a short explanation to a revised manuscript and add conclusions derived from this sample to a revised manuscript.

How do the authors know that they had a representative sample from the hypolimnion at the last sampling date? There is no change in temperature evident and the decline in oxygen concentrations does not reach oxygen concentrations as low as at the earlier time points. Likewise, methane concentration in this sample is not as high as in the other samples from the hypolimnion. Thus, it appears that the sample was not taken at appropriate depth to be comparable with the others.

Answer: During lake overturn the mixed layer (corresponding to epilimnion in our study) increases over time due to gradual cooling and the resulting density change. By January this mixing process is almost complete and the conditions in the remaining "hypolimnion" differ from those earlier in the season. The water at the lake bottom however still shows a small differences of e.g. methane and oxygen concentration to the water above, but the temperature gradient is gone (Figure 1d, Supplementary Fig. 1d). The lake has a max. depth of 16 m and our sample was taken from 15 m, the maximum depth that can be sampled with the niskin bottle without disturbing the sediment. Therefore, if the sample in January is s truly a "hypolimnion" sample is indeed somewhat debatable but refers to the bottom location of the sample as for the other sampling dates. As mentioned above we will better explain the term "hypolimnion sample" in a revised manuscript and describe the caveats that applies to the bottom sample in Jan-
**uary.**

The conclusion about the specific enrichment of well-adapted methanotrophs with particular methane oxidation kinetics (l. 23) is conceivable, but should be drawn more carefully, because it remains unclear whether the observed kinetics are indeed adaptations of particular competitive methanotrophs under oligotrophic conditions, especially with regard to affinity. As only apparent parameters could be estimated, it remains unclear whether the methane monooxygenase of the respective organisms has indeed a higher affinity (lower Km) and is thus more competitive. It should be kept in mind in this context that a low apparent Km is not necessarily a specific adaptation to low methane concentrations, but can be the result of starvation (see Dunfield and Conrad 2000, AEM).

Answer: It is of course true that we only measured the (apparent) kinetic properties of the whole MOB assemblage. Our conclusion is based on the observation that changes in the apparent kinetic properties of the assemblage are accompanied by changes in the community composition and changes in the expression level of pmoABC variants of the assemblage. In this respect, we draw conclusions on an average trend on the community level but we are well aware that there might be additional mechanisms, e.g. on individual species level as well. In a revised manuscript we will formulate the discussion in a more differentiated manner and discuss additional explanations. We will include in the discussion section that changes in Km can be a result of starvation. We also note that Dunfield and Conrad 2000 observed a constant specific affinity (a0), which is in contrast to our results. In our case both Km and a0 change with depth and with time, which may suggest a different mechanism than observed in Dunfield and Conrad (2000). Whether the kinetic differences, assuming they have a genetic basis, have an influence on the competitiveness of a species is indeed a very interesting question. We are currently preparing a manuscript that discusses a modelling approach in which we investigate the interplay of MOB populations consisting of "species" with varying Km and Vmax and their competitiveness in the Rotsee setting in detail.
To determine methane uptake kinetics (Fig. 2), the samples were apparently incubated at the temperatures measured in the epilimnion. However, samples from the hypolimnion encounter much lower temperatures in autumn. How does that affect comparability of the obtained results and conclusions about in situ conditions? This should be taken into account.

Answer: We incubated at an intermediate temperature between epi- and hypolimnion. Our approach was not designed to obtain the most realistic in-situ rates, but to get a valid comparison of the apparent kinetics in the different communities of the sampled water layers. To be able to compare the measured kinetics between epi- and hypolimnion at each given date, we measured them at the same temperature, since temperature influences the solubility of gases (methane). We tried to treat them as similarly as possible. A multi-temperature approach would have been prohibitive for the same logistical reasons discussed in the context of replication above. We propose to make the rationale for our approach more clear in the revision and to mention the caveats that arise from this choice.

Related to this point: Considering that altered temperature and oxygen conditions were used to characterize the methane uptake kinetics in vitro, to what extent can the findings be translated to in situ conditions, considering that these factors can affect the measured Km and Vmax (see the study of Thottathil et al 2019, who report that increasing oxygen concentrations in lake water can reduce maximum methane oxidation rates; doi.org/10.1007/s10533-019-00552-x). Is it conceivable that Vmax in the hypolimnion is underestimated when determining oxidation rates at higher oxygen concentrations in vitro?

Answer: Altered temperature and oxygen may influence methane uptake kinetics, but to our knowledge not much is known about these influences. Influence on methane oxidation rates does not necessarily also mean an influence on apparent methane affinity. Further, a previous study did not find that oxygen has an inhibiting effect in culture studies (Ren et al. 1997) on the other hand some species seem to be microaerophilic
(Rahalkar et al. 2007). In an attempt to measure methane affinities under as standardized conditions as possible we measured the methane oxidation kinetics under similar conditions to the best of our knowledge.

There were additional, methodological reasons to bubble the samples with air prior to incubations. One reason was to remove any background methane. The other reason was that 3H-CH4 is susceptible to some exchange of H with H2S, and purging with air also removes the H2S from the samples. Background exchange with H2S can be also be accounted for by killed controls, but this approach lowers the sensitivity of the method and high sensitivity is very important when measuring methane affinity. One could think about other methods like incubations with 13C-CH4 or 14C-CH4 but these methods likewise do not allow the very high sensitivity which is needed to measure methane oxidation rates at low methane oxidation activities, which is why we chose 3H-CH4 the most sensitive tracer (Bussmann et al. 2015). Another advantage of 3H-CH4 is that due to the high activity of this tracer only small amounts have to be added, which is also very important, because to measure methane affinity the methane oxidation rates have to be measured accurately also at very low methane concentrations.

In a revised manuscript we will add this rationale behind the methane affinity experiments to the methods section. Further, we will add that altered temperature and oxygen concentration as compared to the in-situ conditions may influence the result as a limitation of our study to the discussion section.

I find it very unfortunate that the identification of methanotrophs stops at the level "type la, type lb, type II". The sequence information should provide more detailed information about the identity of the methanotrophs. At least for pmoA comprehensive datasets are available covering besides cultivated strains diverse groups of uncultivated taxa, so that more information could have been extracted here to identify conspicuous taxa.

Answer: In a revised manuscript we will provide an improved classification of pmoA based on available databases (Wen et al. 2016). We note that the pmoABC sequences
are available as fasta files via our data repository.

Specific comments: I. 19 and 291: According to the data in table S1, the difference in Km is 20-fold, not 2 orders of magnitude

Answer: That's correct, we will change this in a revised manuscript.

I. 25: Where in the presented work is it shown or discussed that 90% of the methane are removed? It appears that this is not a conclusion that is derived from the presented work.

Answer: This is a result from our work on the overturn of the same lake one year prior to the present study (Zimmermann et al. 2019), and is based on using mass balance analysis and modelling. We will provide a clear reference for the statement in question.

I. 65: Metagenomic data were used as a basis for the metatranscriptomic data analysis, but are not presented independently; thus, I would not emphasize the metagenomics approach here for the analysis of MOB assemblages.

Answer: In a revised manuscript we will deemphasize the metagenomics part.

I. 73: Five campaigns in autumn 2017 does not appear correct (three samplings in 2017 and one in 2018 according to the presented results)

Answer: Yes, we will change this accordingly in a revised manuscript.

I. 74-75: More measured parameters are given here than presented; harmonize.

Answer: We will harmonize the methods part accordingly in a revised manuscript.

I. 78: I do not find any helpful information about the radio isotope tracer technique in Steinle et al 2015. While the cited references enabled me to understand how methane oxidation rates were determined, they do not allow me to evaluate whether/how this procedure can be used to survey methane oxidation kinetics.

Answer: The radio tracer technique is discussed in detail in Bussmann et al. (2015)
and in the supplements to Steinle et al. (2015). Methane oxidation kinetics are derived from rate measurements and we describe this in detail on line 86 – 108. To our knowledge there is no reference available where methane oxidation kinetics were determined using tritiated methane. In a revised manuscript we will add a reference to Lofton et al. (2014), who derived methane oxidation kinetics from methane oxidation rates that were measured using 14C-CH4. As described in Bussmann et al. 2015, the 3H-CH4 tracer technique is more sensitive than the 14C-CH4 technique and therefore allows shorter incubation times and rate determination at low CH4 concentrations.

**I. 80: How much methane was in this mixture?**

Answer: The specific activity of 3H-CH4 is 0.74 TBq mmol-1. The 200  $\mu$ L of gaseous 3H-CH4/N2 mixture contained 108 pmol 3H-CH4 which corresponds to an activity of 80 kBq. In comparison, the 500  $\mu$ L gas bubble with the lowest concentration of unlabelled methane, contained 17 nmol CH4.

I. 100-106: The authors describe different criteria that were used to identify and eliminate outliers here. Point four states that data points were removed in case less then two replicates remained. According to I. 93, duplicates were prepared. Does that mean that data for a specific methane concentration were lost each time one of the two replicates was identified as outlier? In this context, it is also unclear what Fig. 1 e-h shows. Do the presented data points represent individual measurements or are these mean values of the two replicates? Sometimes, I see two data points at a specific concentration, but sometimes I see only one point. Please clarify.

Answer: In a revised manuscript we will improve clarity and the level of detail of the affected methods sections. We prepared incubations in triplicates except for the first sampling campaign where we only prepared duplicates. Because we only had incubation duplicates for the first sampling date, it was not possible to detect outliers based on  $2\sigma$ . For the incubation duplicates and incubation triplicates we measured each replicate twice and averaged the measurement duplicates. The data points in Fig. 1
e-h represent these averaged values for each individual incubation replicate. In those cases where it seems that there is only one point, the two or three datapoints are so close that they overlap. We will provide additional detail in a revised manuscript to clarify these points. Please also refer to our detailed reply to the other reviewer in this discussion.

It would be valuable to know how many high-quality reads the authors generated per sample in the metagenomic and metatranscriptomic analysis, respectively.

Answer: For metagenomics we generated approx. 31-37 million reads and for metatranscriptomics 26.6 - 34 million reads. In a revised manuscript we will provide this information in the methods section.

I. 157: Why three samples in October; to my understanding there should be one from the epilimnion and one from the hypolimnion per point of time.

Answer: We measured one additional depth in October in between epi- and hypolimnion, which is included in the data repository but is currently not discussed in the manuscript. We did not pursue the intermediate sample in later campaigns since we concluded that focusing our effort on the Epilimnion and Hypolimnion (e.g. continuing with measurement triplicates) would be better. We did however use the metagenomics data from this sample for the assembly. We propose to stick to the two-depth presentation for the main manuscript and figures as it keeps the story focused. However, if it is deemed desirable, we could add information on this additional sample in the supplementary material. We will also provide information on this sample in the methods section.

I. 163: Can a few words be added to describe this custom database? How was it set up? What type of data does it include?

Answer: We prepared three custom databases, one for each gene: pmoA-like, pmoB-like, pmoC-like. We extracted pxmABC, pmoCAB2, pmoCAB from both alpha- and

BGD
gammproteobacterial genomes manually. Note that pmoCAB2 is only known from alphaproteobacterial methanotrophs as yet. We used amino acid sequences with diamond blastp to find the genes in our metagenome and transcriptome. The custom database was especially important for pmoC which was not included in the available prokka databases. A short description will be provided in the manuscript upon revision, and the databases will be made publicly available. See Fig. 1.

I. 202-205, I. 295 and perhaps elsewhere: wording: do the authors refer to Km or a0 here when talking about affinity?

Answer: We refer to Km when talking about "affinity". We refer to a0 when we talk about "specific affinity". We will make sure that these terms are clearly defined on first use, and we think putting the kinetic parameter terms in italics throughout might further help to distinguish them in the text.

I. 204-207: I cannot follow argumentation here. And how do the authors explain that the organisms with the higher Vmax and lower Km disappear in January (Fig. 1h), although they should have a competitive advantage?

Answer: The fact that certain types disappear when they should have a competitive advantage with respect to the methane oxidation kinetics (at least insofar as we can conclude on the traits of individual species from the measurements performed on consortia) indeed leads us to the conclusion that one would need to consider additional traits to explain the observed abundance pattern and its dynamics. As stated on line 205 we believe that there have to be additional important differences in other traits (i.e. temperature optimum or specific adaptations to low or high oxygen concentrations). We already have strong indications from our previous work that these factors are indeed important (Mayr et al. 2020a, 2020b) Nonetheless, we do observe significant differences and changes in the methane oxidation kinetics which are related to the differences and changes in the environment. We propose to improve this section for a revised version by incorporating the information above.

BGD
I. 241: A range of 1 - 40 is a bit outdated. Atmospheric methane oxidizers in soil are meanwhile known to have a0s values with up to  $195 \times 10-12$  L/cell\*h (Tveit et al) and in upland soils, estimates are ranging up to  $800 \times 10-12$  L/cell\*h (Kolb et al 2005; doi:10.1111/j.1462-2920.2005.00791.x)

Answer: Thank you, we will update this in a revised version of the manuscript.

I. 248-250: I find the 25% and 93% values critical here, because huge differences are observed at the individual time points. Especially the 93% value appears to be strongly affected by the huge difference observed in December.

Answer: Yes, we will give more differentiated information upon revision of the manuscript. Individual percentages can be calculated from the individual values for the maximum methane oxidation rate as well as the in-situ methane oxidation rate in Supplementary Table 1. In the epilimnion there might indeed be a trend over time, however all values are below 50%. In the hypolimnion we observe less variation and all values are above 67%. Therefore, we do believe that it is a valid conclusion that MOB in the epilimnion were generally more limited than MOB in the hypolimnion.

I. 254: What do the authors mean with aggregate properties here? What aggregates do they refer to?

Answer: "Thus, the reported kinetics reflect aggregate properties of the respective assemblage." Aggregate was used in the sense of "formed or calculated by the combination of several separate elements", in this case of all the species and individuals in the community, i.e. we meant to express that the reported kinetics reflect the (apparent) properties of the MOB assemblage which is present at the depth and date of sampling. In a revised manuscript we will try to find a less ambiguous term.

I. 256 – 258: It would be very valuable if the described findings could be seen in Figure 3.

Answer: In a revised manuscript we will try to improve this and highlight type Ia Ib and
II in Fig. 3.

I. 292-293: The transcription of genes does not relate to enzyme affinity or apparent Km values; thus, I cannot follow argumentation here.

Answer: The association is only correlational so far, this is true. In a revised manuscript we will tone these statements down or clearly state them as hypotheses.

I. 301: I do not necessarily agree to the term "entirely" in the context with "kinetic traits"; other environmental conditions may have affected the kinetic parameters. Please keep in mind that you can only measure apparent parameters, not enzyme kinetics.

Answer: We will remove the "entirely" there. Yes, it is true that other parameters likely influence kinetic traits. We keep in mind that we do not measure enzyme kinetics but apparent kinetics. Nevertheless, our metatranscriptomics data clearly suggest that pMMO was by far the most expressed methane oxidizing enzyme. Therefore, according to our data most of the methane which is oxidized is oxidized by pMMO and not other enzymes.

I. 303-304: Please note that Methylocapsa gorgona does not possess a second pmoA gene for "high-affinity oxidation" despite being able to live on very low methane concentrations (Tveit et al).

Answer: Yes, we are aware of this interesting study and cite it elsewhere in the manuscript. Methylocapsa gorgona can live and grow on very low methane concentrations, but it does not have a very low apparent Km (4.9  $\mu$ M), but it has a very high specific affinity. But many methanotrophs possess sMMO and pMMO, and sMMO has a lower affinity than pMMO. To our knowledge Methylocapsa gorgona does not have sMMO.

References: The reference list does not allow to differentiate publications (e.g. Mayr et al 2019a, b, c). The reference list lacks information about the year the work has been published and the indices a,b,c.
Answer: Here we followed the author guidelines of the Journal and added the year + a,b,c at the end of the reference.

Mayr, M. J., Zimmermann, M., Guggenheim, C., Brand, A. and Bürgmann, H.: Niche partitioning of methane-oxidizing 429 bacteria along the oxygen-methane counter gradient of stratified lakes, ISME J., doi:10.1038/s41396-019-0515-8, 2019a.

Mayr, M. J., Zimmermann, M., Dey, J., Brand, A. and Bürgmann, H.: Growth and rapid succession of methanotrophs 431 effectively limit methane release during lake overturn, bioRxiv, doi:https://doi.org/10.1101/707836, 2019b.

Mayr, M. J., Zimmermann, M., Dey, J., Wehrli, B. and Bürgmann, H.: Data for: Community methane-oxidation kinetics 433 selected by lake mixing regime [Data set], Eawag Swiss Fed. Inst. Aquat. Sci. Technol., doi:10.25678/0001fa, 2019c.

Figure 1: The axis showing oxygen concentrations should have a more increments.

Answer: We will provide an axis with more increments in a revised manuscript.

Figure 2: explain error bars

Answer: As described in the figure caption we plot 95% confidence intervals as light green and light orange vertical bars. Average values are plotted as dark green and dark orange lines. In a revised manuscript we will improve clarity of the figure caption.

Figure 3: The distinction by color is difficult in plots a1-c1; why not choosing more distinct colors / a broader range of colors per plot? This is of particular importance, as the relative abundances cannot be taken from Table S2 without additional calculations. It is currently impossible to identify type Ib or type II methanotrophs based on the color code and without further invest. However, as pointed out above, it would be even more valuable if more taxonomic information could be provided.

Answer: We have a hard time coming up with a better and still colour-blind proof palette. We decided that taking a clear brightness-based palette with additional colour

BGD
information would be best. If the reviewer or editor have concrete suggestions for better colour palettes, we would greatly appreciate it. We will definitely highlight type Ia, Ib and II in the figure in a revised manuscript. We will provide the supplementary table with the underlying data as comma separated file instead of pdf to make it more accessible. The fasta files of all variants are available in the provided in a data repositors ("Data availability"). In a revised manuscript we can provide a taxonomic classification of pmoA based on available databases.

Table S3: Provide reference for Knief et al 2015.

Answer: We will provide the reference in a revised manuscript.

References

Bussmann, I., A. Matousu, R. Osudar, and S. Mau. 2015. Assessment of the radio 3H-CH4 tracer technique to measure aerobic methane oxidation in the water column. Limnology and Oceanography: Methods 13:312–327.

Lerman, A., and L. Chou. 1995. Physics and chemistry of lakes. Page Lakes: chemistry, geology, physics. [Second ed. Berlin [etc.]ŕ: Springer-Verlag.

Lofton, D. D., S. C. Whalen, and A. E. Hershey. 2014. Effect of temperature on methane dynamics and evaluation of methane oxidation kinetics in shallow Arctic Alaskan lakes. Hydrobiologia 721:209–222.

Mayr, M. J., M. Zimmermann, J. Dey, A. Brand, B. Wehrli, and H. Bürgmann. 2020a. Growth and rapid succession of methanotrophs effectively limit methane release during lake overturn. Communications Biology in press.

Mayr, M. J., M. Zimmermann, C. Guggenheim, A. Brand, and H. Bürgmann. 2020b. Niche partitioning of methane-oxidizing bacteria along the oxygen-methane counter gradient of stratified lakes. The ISME Journal 14:274–287.

Oswald, K., J. S. Graf, S. Littmann, D. Tienken, A. Brand, B. Wehrli, M. Albertsen, H.
Daims, M. Wagner, M. M. M. Kuypers, C. J. Schubert, and J. Milucka. 2017. Crenothrix are major methane consumers in stratified lakes. ISME Journal 11:2124–2140.

Rahalkar, M. C., I. Bussmann, and B. Schink. 2007. Methylosoma difficile gen. nov., sp. nov., a novel methanotroph enriched by gradient cultivation from littoral sediment of Lake Constance. International Journal of Systematic and Evolutionary Microbiology 57:1073–1080.

Ren, T., J. A. Amaral, and R. Knowles. 1997. The response of methane consumption by pure cultures of methanotrophic bacteria to oxygen. Canadian Journal of Microbiol-ogy 43:925–928.

Steinle, L., C. A. Graves, T. Treude, B. Ferré, A. Biastoch, I. Bussmann, C. Berndt, S. Krastel, R. H. James, E. Behrens, C. W. Böning, J. Greinert, C. J. Sapart, M. Scheinert, S. Sommer, M. F. Lehmann, and H. Niemann. 2015. Water column methanotrophy controlled by a rapid oceanographic switch. Nature Geoscience 8:378–382.

Wen, X., S. Yang, and S. Liebner. 2016. Evaluation and update of cutoff values for methanotrophic pmoA gene sequences. Archives of Microbiology 198:629–636.

Yannarell, A. C., and E. W. Triplett. 2004. Within- and between-Lake Variability in the Composition of Bacterioplankton Communities: Investigations Using Multiple Spatial Scales. Applied and Environmental Microbiology 70:214–223.

Zimmermann, M., M. J. Mayr, D. Bouffard, W. Eugster, T. Steinsberger, B. Wehrli, A. Brand, and H. Bürgmann. 2019. Lake overturn as a key driver for methane oxidation. bioRxiv.
| Interactive |
|-------------|
| comment     |
| pmoA-like                                                                                               |              |             |             |             |             |                |                   |              |             |            |
|---------------------------------------------------------------------------------------------------------|--------------|-------------|-------------|-------------|-------------|----------------|-------------------|--------------|-------------|------------|
| >WP_0857                                                                                                | 69855.1 m    | ethane mon  | ooxygenase  | /ammonia ı  | nonooxygei  | nase subuni    | t A [Methyle      | ocystis bryo | phila][pxm/ | v ] |
| >AMK7576                                                                                                | 0.1 methar   | e monooxy   | genase/amn  | nonia mono  | oxygenase   | subunit A [M   | Nethylomor | as denitrifi | cans][pxmA  | J          |
| >AMK7727                                                                                                | 6.1 methar   | e monooxy   | genase/amn  | nonia mono  | oxygenase   | subunit A [M   | Vethylomor        | as denitrifi | cans][pmoA  | ]          |
| >WP_0857                                                                                                | 72041.1 m    | ethane mon  | ooxygenase  | /ammonia ı  | nonooxygei  | nase subuni    | t A [Methyle      | ocystis bryo | phila][pmo/ | \2]        |
| >WP_085772257.1 methane monooxygenase/ammonia monooxygenase subunit A [Methylocystis bryophila][pmoA1]  |              |             |             |             |             |                |                   | \1]          |             |            |
| >WP_085772257.1 methane monooxygenase/ammonia monooxygenase subunit A [Methylocystis bryophila][pmoA1b] |              |             |             |             |             |                |                   |              |             |            |
| >WP_0331                                                                                                | 57542.1 M    | ULTISPECIES | : ammonia I | monooxyge   | nase [Meth  | vlomonas][p    | oxmA]             |              |             |            |
| >WP_0331                                                                                                | 59227.1 m    | ethane mon  | ooxygenase  | /ammonia ı  | nonooxygei  | nase subuni    | t A [Methyle      | omonas sp.   | LW13][pma   | A]         |
| >0000238                                                                                                | 5.1 Particu  | ate methan  | e monooxyį  | genase beta | subunit [Ch | romatiales     | bacterium l       | JSCg_Taylo   | r][pmoA_US  | Cg]        |
| >CCJ08278                                                                                               | .1 Particula | te methane  | monooxyge   | nase subun  | it A [Methy | locystis sp. ! | SC2]              |              |             |            |
| >CCJ08985                                                                                               | .1 Particula | te methane  | monooxyge   | nase subun  | it A [Methy | locystis sp. 1 | SC2]              |              |             |            |
|                                                                                                         |              |             |             |             |             |                |                   |              |             |            |

**pmoB-like**

>ACE95891.2 particulate methane monooxygenase subunit B-like protein [Methylomonas sp. LW13] [pxmB]
>ACE95892.2 particulate methane monooxygenase subunit B-like protein [Methylomonas methanica] [pxmB]
>WP\_085769854.1 methane monooxygenase/ammonia monooxygenase subunit B [Methylomonas denitrificans][pxmB]
>AMK7277.1 methane monooxygenase/ammonia monooxygenase subunit B [Methylomonas denitrificans][pxmB]
>MWP\_085779042.1 methane monooxygenase/ammonia monooxygenase subunit B [Methylomonas denitrificans][pxmB]
>MWP\_085772042.1 methane monooxygenase/ammonia monooxygenase subunit B [Methylocystis bryophila][pmoB]
>WP\_085772258.1 methane monooxygenase/ammonia monooxygenase subunit B [Methylocystis bryophila][pmoB1]
>WP\_085772258.1 methane monooxygenase/ammonia monooxygenase subunit B [Methylocystis bryophila][pmoB1]
>WP\_033157541.1 MULTISPECIES: methane monooxygenase/ammonia monooxygenase subunit B [Methylocystis bryophila][pmoB1]
>WP\_0331572258.1 methane monooxygenase/ammonia monooxygenase subunit B [Methylocystis bryophila][pmoB1]
>WP\_033157341.1 MULTISPECIES: methane monooxygenase/ammonia monooxygenase subunit B [Methylocystis bryophila][pmoB1]
>WP\_0331537541.1 MULTISPECIES: methane monooxygenase/ammonia monooxygenase subunit B [Methylocystis bryophila][pmoB1]
>OO002386.1 Particulate methane monooxygenase alpha subunit [Chromatiales bacterium USCg\_Taylor][pmoB\_g]
>OO002386.1 Particulate methane monooxygenase alpha subunit [Chromatiales bacterium USCg\_Taylor][pmoB\_g]

**pmoC-like**

| >WP_085769853.1 methane monooxygenase/ammonia monooxygenase subunit C [Methylocystis bryophila] [pxmC]  |  |  |  |  |  |  |  |
|---------------------------------------------------------------------------------------------------------|--|--|--|--|--|--|--|
| >AMK75758.1 methane monooxygenase/ammonia monooxygenase subunit C [Methylomonas denitrificans] [pxmC]   |  |  |  |  |  |  |  |
| >AMK77275.1 methane monooxygenase/ammonia monooxygenase subunit C [Methylomonas denitrificans] [pmoC]   |  |  |  |  |  |  |  |
| >WP_085772039.1 methane monooxygenase/ammonia monooxygenase subunit C [Methylocystis bryophila][pmoC2]  |  |  |  |  |  |  |  |
| >WP_085771887.1 methane monooxygenase/ammonia monooxygenase subunit C [Methylocystis bryophila][pmoC1]  |  |  |  |  |  |  |  |
| >WP_085771887.1 methane monooxygenase/ammonia monooxygenase subunit C [Methylocystis bryophila][pmoC1b] |  |  |  |  |  |  |  |
| >WP_033157540.1 MULTISPECIES: methane monooxygenase/ammonia monooxygenase subunit C [Methylomonas][p:   |  |  |  |  |  |  |  |
| >WP_033159228.1 MULTISPECIES: methane monooxygenase/ammonia monooxygenase subunit C [Methylomonas][pm   |  |  |  |  |  |  |  |
| >00002384.1 hypothetical protein USCGTAYLOR_01391 [Chromatiales bacterium USCg_Taylor][pmoC_g]          |  |  |  |  |  |  |  |
| >CCJ06178.1 Methane monooxygenase subunit C [Methylocystis sp. SC2] [pmoC_cis]                          |  |  |  |  |  |  |  |
| >CCJ06303.1 Particulate Methane Monooxigenase subunit C [Methylocystis sp. SC2] [pmoC_cis]              |  |  |  |  |  |  |  |
| >CCJ08277.1 Particulate methane monooxigenase subunit C [Methylocystis sp. SC2][pmoC]                   |  |  |  |  |  |  |  |
| >CCJ08984.1 Particulate Methane Monooxigenase subunit C [Methylocystis sp. SC2][pmoC]                   |  |  |  |  |  |  |  |
| >CCJ05653.1 Particulate methane monooxigenase subunit C, PmoC2 protein [Methylocystis sp. SC2] [pmoC2]  |  |  |  |  |  |  |  |
| >AMK77275.1 methane monooxygenase/ammonia monooxygenase subunit C [Methylomonas denitrificans]          |  |  |  |  |  |  |  |

Fig. 1.

**BGD**

Interactive

comment

**Reply to Reviewer 2:**

Mayr, Zimmermann and colleagues studied the methane oxidation kinetics in the epiand hypofimions of a eutrophic lake during autumn/winter lake overturn and report changing methane uptake kinetics. Likewise, changes in punc2, A and B gene expression profiles were observed, indicating adjustments in the active methanotrophic community in dependence on methane availability. I see value in the presented work, but also limitations and open questions that the uotid have to be carried.

Answer: Thank you for your valuable comments. In the following we try to answer and clarify all open questions and limitations raised.

First of all, I request the authors to point out that they measured apparent methane oxidation kinetics. This should be clearly indicated throughout the manuscript.

Answer: We agree. In a revised manuscript we would change to "apparent methane oxidation kinetics" throughout the text.

In this study, all conclusions are derived from 1 - 21 of water per sample, taken at C1 four different time points of the epillminon and hyopkinnion, respectively. However, no repitate samples were taken per layer, 11 find this hardly acceptable. How representative are the findings of 11 water for the whole stratification layer of a lake? Can the authors be sure that the difference they see are indeer related to the respective water bodies? Alteraby the molecular analysis of a second filter, which is a methodological repixel that with redshord the too sample, shows some differences (Fig. 3). Thus, 11 find it it negrely impossible to relate differences in the active methanotrophic community to stratification, teayoing anything about the biological variation within a layer. The repeated measurements over time provide some evidence, but do not solve this issue when it comes to minor differences between specific.

Answer: Horizontal mixing in lakes is strong (Lerman and Chou, 1995, page 86), especially in stratified lakes and horizontal variation within a lake (excluding near-shore water or parts with limited water exchange) is expected to be small (Yannarell and Tripiet 2004) in comparison with the verticeal variation during stratification and temporal variation of the mixed layer. Taking one profile, usually close to the deepset point is therefore common practice for studies in smaller lakes or date surveys (Owand et al. 2017). Smaller volume to redpet the the volumes representativeness.

In our cartier study on Rotsee we investigated the general population structure around the oxycline and later the community composition change over time during lake overtrum in considerable detail. We found that especially the mixed surface layer is a very homogeneous environment considering both environmental conditions and the mechanotroph assemblage (May et al. 2020a). Therefore, we think that the epiliminon measurems are indeed representative for the mixed layer (epiliminon in this study) and only very small variabilities would be expected from multiple measurements. The focus of this study was not on variability within the mixed layer or within hypolimino, but rather to demonstrate that differences in kinetic parameters occur at all, in contrasting microbial communities within one lake which has not been demonstrated before. Therefore, although we did not study the variation within a layer in this study, we had considerable previous knowledge on the same lake from the years before and designed the study on this basis.

The hypolimnion during stratification by definition shows very different conditions to the epilimion, but gradients of mehane and other parameters can be observed within the hypolimnion. Therefore, whereas the hypolimnion is less homogeneous than the epiliminon, the difference to the epiliminoin is clear based on both, the environmental conditions (orders of magnitude higher mehane concentrations) and a different mehanotrophic assemblage (this tudy and Mayr et al. 2020a, 2020b.)

Regarding the request for (biological) replication of the kinetic measurements, we have to note that, although measuring different depths and or replicates from the layers would be favourable, we had to adjust the amount of incubations to our handling limit of incubations. We decided to improve the replication for the respective depths to have enough data points to determine the affinity of the respective depth (sec our answer on other elimitation).
Fig. 2.

---

## Referee Comment (RC3) · Anonymous Referee #3 · 9 Mar 2020

Dear Authors,

This manuscript describes the study of methane oxidation (MOX) during lake overturn in Lake Rotsee in Switzerland. You combine measurements of MOX kinetics with meta-transcriptomic analyses of methane monooxygenase genes and report differences between epi-and hypolimnion during stratification and a convergence of MOX kinetics and gene expression during lake water mixing. You conclude that methane oxidizers with well-adapted kinetics occupy distinct niches in stratified lakes.

While I think the report of kinetic parameter of methane oxidation is of great relevance, however, I found that the manuscript suffers from a lack of clarity and oversimplifications. Most importantly, it's unclear how the central conclusion, that well-adapted methanotrophs inhabit niches depending on methane availability (in hypo- and epilimnion), is reached. Wouldn't a match between in situ CH4 concentration and Km (not normalized per cell) be a stronger indication of such an adaptation?

I also believe a better use of the metatranscriptomic data could help to strengthen this point. A finer taxonomic resolution based on the pmoCAB genes and a more quantitative characterization of the community turnover should be possible – and could help to make the point that indeed there are distinct populations of MOB that are adapted to in situ CH4 concentration. Accordingly, I think that Figure 3 a1-c1 is not the ideal way to convey this important point. Maybe a combination of SI Fig. 2 (which I think shows quite nicely the convergence towards similar gene expression patterns in January, with a Figure showing the taxonomic composition of MOB during lake overturn would be a better choice.

Moreover, I was somewhat irritated by the rather vague description of the environmental conditions during lake overturn. The traditional definition of lake stratification and hence the difference between epi- and hypolimnion based on temperature rather than oxygen. And while the manuscript addresses MOX during lake overturn, you refer to the oxycline for sampling. I understand that the temperature profiles shown in SI Fig 1 may not be as clear as the oxygen profiles shown in Fig. 1 – but I would advise to show all profiles (also conductivity which should explain the inverse stratification pattern in December) and to be very clear with the definition of overturn, thermos- and oxycline.

Finally, given the relatively low number of samples and the fact that the pattern was (only) observed in Lake Rotsee, I think the manuscript should be thoroughly rewritten to make clear that this may reflect a specific situation in the (relatively eutrophic) Lake Rotsee. Also, there are several cases of speculation or exaggerated extrapolation, which should be avoided.

Please also consider specific comments below:

L 11 In freshwater lakes. . . so, this excludes saline lakes? Consider removing "freshwater"

L 14 we tested the hypothesis that methanotroph assemblages in a seasonally stratified lake. . .

L 18 consider a brief explanation of the meaning "half-saturation constant" here

L19 . . .Km differed by two orders of magnitude – but in the results it seems that they differed between 15 and 0.7 uM (a factor of ∼20)

L 25 . . .90% of what?

L28 can you talk about a climate IMPACT of lacustrine systems?

L 31 anoxic habitats. . . .. In the oxygen-depleted hypolimnion. . . repetitive

L 47 kinetic traits . . .. Use kinetic parameter instead (see L 48)

L 58 . . . Lake Rotsee. . .

L 63 ex situ consider replacing with "laboratory incubations"

L 73 four or five campaigns?

L 77 and onward. Please provide more detail on this method including how the killed controls were treated.

L 91 how were Schott bottles sealed air-tight?

L 110 we determined the in-situ MOX rate . . . in duplicate ex-situ incubations. . . .. Confusing, please rewrite.

L 161 an 167 reads shorter than 400 or 300 bp were removed?

L 183 aerobic methane oxidation likely contributed to this oxygen depletion in the epilimnion. This seems very speculative for me. Could a back of the envelope calculation, e.g. knowing the volume and CH4 concentration in the hypolimnion and the stoichiometry of MOX be used to support this speculation?

L 228 critical phase – critical for what?

L 233 specific affinity towards methane. . . unclear what is meant here.

L 235 was the convergence only driven by changes in kinetic parameter in the epilimnion (or also in the hypolimnion as seems apparent from Fig. 2 a)

L 289 remove "as in many other stratified lakes" – too speculative (or include references, but I would not advise so in the conclusion part)

L 295 adaptation to oligotrophic conditions – Lake Rotsee can not be considered oligotrophic

L 298 transport of methane into the epilimnion provided and advantage for fast-growing MOB over slower competitors. This is not shown (at least in this manuscript) and should be removed.

---

## Author Comment (AC4) · 31 Mar 2020

Reply to Reviewer 3:

This manuscript describes the study of methane oxidation (MOX) during lake overturn in Lake Rotsee in Switzerland. You combine measurements of MOX kinetics with meta-transcriptomic analyses of methane monooxygenase genes and report differences between epi-and hypolimnion during stratification and a convergence of MOX kinetics and gene expression during lake water mixing. You conclude that methane oxidizers with well-adapted kinetics occupy distinct niches in stratified lakes.

Answer: Thank you for your valuable comments. In the following, we address all open questions and limitations raised.

While I think the report of kinetic parameter of methane oxidation is of great relevance, however, I found that the manuscript suffers from a lack of clarity and over-simplifications. Most importantly, it's unclear how the central conclusion, that welladapted methanotrophs inhabit niches depending on methane availability (in hypo- and epilimnion), is reached. Wouldn't a match between in situ CH4 concentration and Km (not normalized per cell) be a stronger indication of such an adaptation?

Answer: Thank you for your positive assessment of the relevance to report kinetic parameters.

We do indeed find a higher affinity (low Km) in the low-methane epilimnion compared to the methane rich hypolimnion as long as stratification is present, and this finding is a key part of our argument (see e.g. abstract). That the Km values do not match the in-situ methane concentrations is however not unexpected. In the mixed layer and under the assumption of a steady-state, the flux of methane from the hypolimnion is balanced with the (non-normalized) methane oxidation rate. Under these conditions, the in-situ methane concentration does depend on the half-saturation constant (Km) but should be lower than Km. Per definition, the half-saturation constant is the substrate concentration where the growth rate is half the maximum growth rate. Even if the growth rate is only half the maximum growth rate, microbial methane oxidation continues, and methane concentrations decrease to values below Km. We added the calculations behind this argument as a supplement to this author reply and we will consider adding this rationale and the calculations to the revised manuscript. In the hypolimnion, oxygen likely becomes limiting and, as a consequence, methane concentrations can be much higher than the half-saturation constant.

In the revised manuscript, we will improve the clarity of our central conclusion and avoid over-simplification. We can incorporate the above points into the discussion for clarity.

I also believe a better use of the metatranscriptomic data could help to strengthen this point. A finer taxonomic resolution based on the pmoCAB genes and a more quantitative characterization of the community turnover should be possible – and could help to make the point that indeed there are distinct populations of MOB that are adapted to in situ CH4 concentration. Accordingly, I think that Figure 3 a1-c1 is not the ideal way to convey this important point. Maybe a combination of SI Fig. 2 (which I think shows quite nicely the convergence towards similar gene expression patterns in January, with a Figure showing the taxonomic composition of MOB during lake overturn would be a better choice.

Answer: Yes, we agree. We will provide a finer taxonomic resolution for pmoA in the revised manuscript. pmoC and pmoB are less indicative for taxonomic classification because they are not usually sequenced and used for this purpose. Yes, we will consider using the SI Fig. 2 instead, which shows the community turnover more clearly in a revised manuscript.

  Moreover, I was somewhat irritated by the rather vague description of the environmental conditions during lake overturn. The traditional definition of lake stratification and hence the difference between epi- and hypolimnion based on temperature rather than oxygen. And while the manuscript addresses MOX during lake overturn, you refer to the oxycline for sampling. I understand that the temperature profiles shown in SI Fig 1 may not be as clear as the oxygen profiles shown in Fig. 1 – but I would advise to show all profiles (also conductivity which should explain the inverse stratification pattern in December) and to be very clear with the definition of overturn, thermos- and oxycline.

Answer: We will add the temperature and conductivity profiles to Figure 1 in the revised manuscript and define overturn, thermo- and oxycline more clearly.

Finally, given the relatively low number of samples and the fact that the pattern was (only) observed in Lake Rotsee, I think the manuscript should be thoroughly rewritten

to make clear that this may reflect a specific situation in the (relatively eutrophic) Lake Rotsee. Also, there are several cases of speculation or exaggerated extrapolation, which should be avoided.

Answer: The effort of obtaining the time resolved data for two water layers was very substantial (described in the answers to Reviewer 1 and 2). Nevertheless, we agree that we only have a limited number of samples from a single lake. We will highlight this more clearly in a revised manuscript and explicitly state that further investigations in other lakes will be required to confirm our findings.

L 11 In freshwater lakes. . . so, this excludes saline lakes? Consider removing "freshwater"

Answer: We will remove freshwater here and mention it in Line 14: "in a seasonally stratified freshwater lake".

L 14 we tested the hypothesis that methanotroph assemblages in a seasonally stratified lake. . .

Answer: In the revised manuscript we will make sure that we only studied a single lake and will specifically discuss limitations of the transferability of our results.

L 18 consider a brief explanation of the meaning "half-saturation constant" here

Answer: We will add a brief definition of "half-saturation constant" in the revised manuscript.

L19 . . .Km differed by two orders of magnitude – but in the results it seems that they differed between 15 and 0.7 uM (a factor of $\sim$20)

Answer: We will correct this in the revised manuscript.

L 25 . . .90% of what?

Answer: 90% of the methane that is transferred to the epilimnion during lake overturn

is oxidized. Since this is a result from an earlier study we will rephrase this sentence in the revised manuscript.

L28 can you talk about a climate IMPACT of lacustrine systems?

Answer: Yes, we think so. According to DelSontro et al (2018), lacustrine systems emit an equivalent of about 20% of the global fossil fuel $CO_2$ emissions, and methane contributes approx. 75% to this. Continuing eutrophication of lacustrine systems will most likely further increase emissions by another 30-90% (DelSontro et al. 2018, Beaulieu et al. 2019) We suggest to clarify: "Methane is a major contributor to the climate impact of greenhouse gas emissions from lakes".

L 31 anoxic habitats.... In the oxygen-depleted hypolimnion... repetitive

Answer: We suggest to remove "anoxic habitats" in the revised manuscript.

L 47 kinetic traits... Use kinetic parameter instead (see L 48)

Answer: We will harmonize the use of "kinetic traits" and "kinetic parameter" in the revised manuscript.

L 58 ...Lake Rotsee...

Answer: We will remove the "a" written before "Lake Rotsee".

L 63 ex situ consider replacing with "laboratory incubations"

Answer: Yes, we will replace this in the revised manuscript.

L 73 four or five campaigns?

Answer: Thanks, four campaigns. We will correct this in the revised manuscript.

L 77 and onward. Please provide more detail on this method including how the killed controls were treated.

Answer: We will provide more detail in the revised manuscript. The killed controls were

treated in the exact same way as the samples with the exception that we added 1 mL of $ZnCl_2$ (50% v/w) to stop biological activity right after we filled the serum vial with the sample. The average water fraction radioactivity of the killed controls were used as background radioactivity in the outlier detection procedure.

L 91 how were Schott bottles sealed air-tight?

Answer: The Schott bottles were not sealed air-tight. Since gasses were stripped from the samples anyway for these analyses, this was not necessary. For determinations of methane concentration and "in-situ" rates, we directly filled samples into serum vials and those were sealed air-tight (see sections 2.4, 2.5).

L 110 we determined the in-situ MOX rate... in duplicate ex-situ incubations... Confusing, please rewrite.

Answer: We suggest to remove "in-situ" and change ex-situ incubations to laboratory incubations in the revised manuscript.

L 161 an 167 reads shorter than 400 or 300 bp were removed?

Answer: We first used a general approach to identify genes (prodigal) removing <400bp genes. During targeted gene identification with prokka and diamond again shorter gene pieces of MMO were identified and we removed short gene fragments <300bp. We will consider harmonizing the base pair cut-off in a revised manuscript.

  L 183 aerobic methane oxidation likely contributed to this oxygen depletion in the epilimnion. This seems very speculative for me. Could a back of the envelope calculation, e.g. knowing the volume and CH4 concentration in the hypolimnion and the stoichiometry of MOX be used to support this speculation?

Answer: We assume that the methane oxidation, leading to the oxygen depletion, mainly occurs in the epilimnion itself. This methane oxidation can occur despite the low methane concentration and is fuelled by the flux of methane into the epilimnion. The stoichiometry of microbial methane oxidation is: $\tilde{a}\breve{A}\acute{U}CH\tilde{a}\breve{A}\mathring{U}\_4+(2-y) O\_2 \rightarrow (1-y) \tilde{a}\breve{A}\acute{U}$-

$CO_2 + y \cdot CH_2O + M + (2-y) H_2O$ where y is the carbon use efficiency. Based on theoretical considerations and experimental data, a carbon use efficiency of 0.4 has been reported (Leak & Dalton 1986). This means that per mole of methane 1.6 moles of oxygen are used. The mixed layer depths for the four sampling campaigns are roughly 6, 10, 12 and 14m, which results in mixed layer volumes of 2.5, 3.7, 4.1 and 4.3 GL in Lake Rotsee. Multiplying the measured methane oxidation rates in the epilimnion with these volumes results in a total methane oxidation of 600, 11560, 11800 and 200 mol d-1. Integrated over the time period of the four campaigns, this results in a total of 0.66 Mmol of methane that were oxidized during this time. This corresponds to a removal of 1.1 Mmol of oxygen from the epilimnion. In an average volume of the mixed layer of 3.7 GL with an initial concentration of 340 $\mu$M (10.9 mg L-1) of oxygen, this would reduce the oxygen concentration by 180 $\mu$M to 160 $\mu$M or to about 5 mg L-1. Note that possible oxygen production and exchange with the atmosphere are not included here. We will consider adding this rationale to the revised manuscript.

L 228 critical phase – critical for what?

Answer: During this time, methane that has accumulated in the hypolimnion is rapidly transported to the surface and is potentially released to the atmosphere – i.e. critical for potential outgassing to the atmosphere and thus for climate relevance. We'll clarify this in the revised manuscript.

L 233 specific affinity towards methane. . . unclear what is meant here.

Answer: We wanted to specify that we mean the specific affinity for methane and not for any other nutrient and will make this clear in the revised manuscript.

L 235 was the convergence only driven by changes in kinetic parameter in the epilimnion (or also in the hypolimnion as seems apparent from Fig. 2 a)

Answer: The convergence was driven by changes in kinetic parameter in the epilimnion and the hypolimnion. We'll clarify this in a revised version of the manuscript.

L 289 remove "as in many other stratified lakes" – too speculative (or include references, but I would not advise so in the conclusion part)

Answer: We will rephrase this sentence in the revised manuscript and provide a reference. That low methane concentrations are found in the epilimnion and higher methane concentrations in the hypolimnion is very common, especially in small lakes e.g. (Bastviken et al. 2004, Borrel et al. 2011), which can vary seasonally like in Rotsee or in permanently stratified lakes elevated methane concentrations can persist over many decades. Continuing eutrophication of lakes and lack of recovery of eutrophic lakes will likely increase the number of lakes with anoxic methane-rich bottom waters in future (Jenny et al. 2016, Beaulieu et al. 2019).

We will tone down the generalized claims elsewhere in the manuscript in order to put our results into context without overstating generalizability.

L 295 adaptation to oligotrophic conditions – Lake Rotsee can not be considered oligotrophic

Answer: What we mean is adaptation to low methane availability, we will change the term accordingly.

L 298 transport of methane into the epilimnion provided and advantage for fast-growing MOB over slower competitors. This is not shown (at least in this manuscript) and should be removed.

Answer: We have provided evidence for the dynamic adaptation in (Mayr et al. 2020), but the reviewer is correct that the phrasing here is misleading and while our data here is in line with the previous investigation the conclusion cannot be made from the data in this paper. We will remove or rephrase this sentence in the revised manuscript.

References:

Bastviken, D., J. Cole, M. Pace, and L. Tranvik. 2004. Methane emissions from lakes: Dependence of lake characteristics, two regional assessments, and a global estimate.

Global Biogeochemical Cycles 18:1–12.

Beaulieu, J. J., T. DelSontro, and J. A. Downing. 2019. Eutrophication will increase methane emissions from lakes and impounds during the 21st century. Nature Communications 10:1375.

Borrel, G., D. Jézéquel, C. Biderre-Petit, N. Morel-Desrosiers, J. P. Morel, P. Peyret, G. Fonty, and A. C. Lehours. 2011. Production and consumption of methane in freshwater lake ecosystems. Research in Microbiology 162:833–847.

DelSontro, T., J. J. Beaulieu, and J. A. Downing. 2018. Greenhouse gas emissions from lakes and impoundments: Upscaling in the face of global change. Limnology and Oceanography Letters 3:64–75.

Jenny, J.-P., P. Francus, A. Normandeau, F. Lapointe, M. E. Perga, A. Ojala, A. Schimmelmann, and B. Zolitschka. 2016. Global spread of hypoxia in freshwater ecosystems during the last three centuries is caused by rising local human pressure. Global Change Biology 22:1481–1489.

Mayr, M. J., M. Zimmermann, J. Dey, A. Brand, B. Wehrli, and H. Bürgmann. 2020. Growth and rapid succession of methanotrophs effectively limit methane release during lake overturn. Communications Biology 3:108.

Please also note the supplement to this comment:
https://www.biogeosciences-discuss.net/bg-2019-482/bg-2019-482-AC4-supplement.pdf

---

## Author Comment (AC5) · 31 Mar 2020

- 1 Supplementary Material: Lake mixing regime selects methane-
- 2 oxidation kinetics of the methanotroph assemblage

**3 Dependency of the steady-state methane concentration in the epilimnion 4 on the half-saturation constant**

5 We consider a well-mixed epilimnion that contains a microbial species B in  $\mu$ M C 6 that grows on a single carbon substrate S in  $\mu$ M C according to the Monod growth 7 kinetics. Furthermore, we assume that the substrate is released from the sediment 8 of the lake and that the concentration gradient from the sediment to the epilimnion 9 has reached a steady state. In this case, we have a constant volumetric flux  $F_S$  in  $\mu$ M 10 C d-1 into the epilimnion.

Without microbial growth, exchange with the atmosphere and substrateconsumption, the change of the substrate concentration in the epilimnion is simply

$$\frac{dS}{dt} = F_S \tag{1}$$

13 According to the Monod kinetics, the growth rate r in d-1 of the microbial species is:

$$r = V_{max} \frac{S}{S + K_M} \tag{2}$$

14 where  $V_{max}$  is the maximum growth rate in d-1 and  $K_M$  the half-saturation constant 15 in  $\mu$ M.

16 Considering a mortality rate m in d-1 and a substrate use efficiency y, the temporal 17 change of the bacterial species is:

$$\frac{dB}{dt} = (y \cdot r - m)B \tag{3}$$

18 Due to the growth of the bacterial species, the substrate concentration in the well-19 mixed epilimnion decreases with a rate of -rB and the complete system of 20 differential equations is:

$$\frac{dS}{dt} = F_S - rB$$

$$\frac{dB}{dt} = (y \cdot r - m)B$$
(4)

**21 Steady-state consideration 1**

A complete steady state is reached if both the concentration of the substrate as wellas the biomass of the microbial species remain constant over time:

$$\frac{dS}{dt} = 0 = F_S - rB$$

$$\frac{dB}{dt} = 0 = (y \cdot r - m)B$$
(5)

By rearranging the second equation, we get an equation for the equilibriumsubstrate concentration:

$$S^{eq} = \frac{K_M}{y \cdot V_{max}/m - 1} \tag{6}$$

According to equation 6, the equilibrium substrate concentration is indeed a function of the half-saturation constant  $K_M$ . However, the equilibrium concentration is not necessarily equal to the half-saturation constant. Depending on the ratio  $y \cdot V_{max}/m$  the equilibrium concentration can be lower or higher than the halfsaturation constant.

The equilibrium concentration is smaller than the half-saturation constant for  $y \cdot V_{max} > 2m$  and is higher than the half-saturation for  $m < y \cdot V_{max} < 2m$ . Because the maximum growth rate is likely higher than twice the mortality rate, the equilibrium concentration is smaller than the half-saturation constant.

35 Estimated equilibrium concentration in the epilimnion of Lake Rotsee: In the 36 epilimnion, we measured an average  $K_M$  of 1.8  $\mu$ M. Based on literature values, we 37 can assume a substrate use efficiency of 0.4 (Leak & Dalton, 1986) and a mortality of 38 0.022 day-1 (Roslev & King, 1995). In the epilimnion, we measured an average 39 maximum methane oxidation rate of about 4 fmole h-1 cell-1. Assuming a cellular 40 carbon content of 0.42 pmole C cell-1 (Oswald et al., 2015; Posch et al., 2001; 41 Romanova & Sazhin, 2010) this converts to a maximum methane oxidation rate of 42 0.2 day-1.

43 With this, we would expect an equilibrium methane concentration of:

$$S^{eq} = \frac{1.8 \,\mu\text{M}}{0.4 \cdot 0.2 \,\text{day}^{-1}/0.022 \,\text{day}^{-1} - 1} = 0.7 \,\mu\text{M}$$
(7)

The highest measured maximum methane oxidation rate was 8.4 fmole  $h^{-1}$  cell-1, which converts to a methane oxidation rate of 0.48 day-1. This higher rate woul result in an equilibrium concentration of about 0.2  $\mu$ M.

These estimates should be treated with caution because all the values used for the calculation (in particular mortality and cellular carbon content) are fraught with considerable uncertainty. Nonetheless, it demonstrates that the expected methane concentration is lower than the half-saturation constant  $K_M$ .

**51 Steady-state consideration 2**

In an alternative approach we assume steady-state for the substrate concentrationbut not for the biomass:

$$\frac{dS}{dt} = 0 = F_S - rB$$

$$\frac{dB}{dt} = (y \cdot r - m)B$$
(8)

54 By rearranging the first equation, we get an equilibrium substrate concentration of:

$$S^{eq} = \frac{F_S \cdot K_M}{B \cdot V_{max} - F_S} \tag{9}$$

55 Estimated equilibrium concentration in the epilimnion of Lake Rotsee: In the epilimnion, we measured an average  $K_M$  of 1.8  $\mu$ M. With a cellular carbon content of 56 0.42 pmole C cell-1 (Oswald et al., 2015; Posch et al., 2001; Romanova & Sazhin, 57 2010) and average cell numbers of  $0.1 \times 10^5 - 2.5 \times 10^5$  cells mL-1, we estimate average 58 biomass of 4 – 105  $\mu$ M. In the epilimnion we measured an average maximum 59 methane oxidation rate of about 4 fmole h-1 cell-1. Using the above cellular carbon 60 61 content, this converts to a maximum methane oxidation rate of 0.2 day-1. Based on model work (Zimmermann et al., 2019), the median volumetric flux of methane into 62 63 the epilimnion is 5  $\mu$ M day-1.

64

65 Inserting these values into equation 9 results in an equilibrium methane 66 concentration of  $0.6 - 1.5 \mu$ M, depending on the amount of biomass (lower 67 equilibrium concentrations with higher biomass). Again, the expected methane 68 concentration tends to be lower than the half-saturation constant  $K_M$ .

69 70

**71 **References**

- Leak, D. J., & Dalton, H. (1986). Growth yields of methanotrophs 1. Effect of copper
  on the energetics of methane oxidation. *Applied Microbiology and Biotechnology*, 23(6), 470–476.
- Oswald, K., Milucka, J., Brand, A., Littmann, S., Wehrli, B., Kuypers, M. M. M., &
   Schubert, C. J. (2015). Light-Dependent Aerobic Methane Oxidation Reduces
- 77 Methane Emissions from Seasonally Stratified Lakes. *Plos One, 10*(7).
- Posch, T., Loferer-Krößbacher, M., Gao, G., Alfreider, A., Pernthaler, J., & Psenner, R.
   (2001). Precision of bacterioplankton biomass determination: a comparison of
   two fluorescent dyes, and of allometric and linear volume-to-carbon conversion
   factors. Aquatic Microbial Ecology, 25(1), 55–63.
- Romanova, N. D., & Sazhin, A. F. (2010). Relationships between the cell volume and
   the carbon content of bacteria. *Oceanology*, *50*(4), 522–530.
- Roslev, P., & King, G. M. (1995). Aerobic and anaerobic starvation metabolism in
   methanotrophic bacteria. *Applied and Environmental Microbiology*, 61(4),

**86 1563–1570.**

- Zimmermann, M., Mayr, M. J., Bouffard, D., Eugster, W., Steinsberger, T., Wehrli, B.,
  ... Bürgmann, H. (2019). Lake overturn as a key driver for methane oxidation.
- 89 BioRxiv.

90

---

## Author Response (AR1)

Eawag 6047 Kastanienbaum Management

Seestrasse 79 Surface Waters - Research and Switzerland Dr. Magdalena Mayr www.eawag.ch Magdalena.mayr@eawag.ch

Biogeosciences Editorial board

Lucerne, 15 June 2020

Dear Prof. Dr. Battin.

Thank you for the positive assessment of our manuscript "Lake mixing regime selects apparent methaneoxidation kinetics of the methanotroph assemblage" (bg-2019-482). On behalf of all authors, I am submitting a new version of the manuscript with the requested major revisions.

We implemented all major and minor comments of the three reviewers, as outlined previously in our author replies to the reviewer comments. I have included our final replies to the reviewer comments within this document. Further, we provide the marked-up version of the manuscript.

If there are further changes required, we are happy to hear from you.

Yours sincerely,

Magdalena Mayr (On behalf of all authors: Magdalena J. Mayr, Matthias Zimmermann, Jason Dey, Bernhard Wehrli and Helmut Bürgmann)

**Major changes:**

We revised and improved Fig. 1 according to the reviewer comments by adding the measured temperature profiles. We also revised Fig. 3 according to the suggestions of reviewer 2 and 3. We now show an ordination in the main figure. Further, we provide taxonomic classification for the *pmoA* sequences (Fig. 3) as requested by reviewer 2 and 3 - and we also included a phylogenetic pmoA tree in the supplementary material (Supplementary Fig. 2).

In the revised manuscript we discuss in detail the relation between methane affinity, maximum methane oxidation rates, methanotroph biomass and their influence on the methane concentration in the epilimnion. To do so we provide the underlying rational and calculations in the supplementary material. Further, we now discuss the influence of methane oxidation on the oxygen concentration in the epilimnion and provide an estimate.

Since there were several comments regarding the nature of the lake mixing regime and the characteristics of epi- and hypolimnion, we added a general paragraph describing the autumn overturn characteristics.

Further we added additional information to the methods section, as requested, and provide the custom databases used for finding pmoABC variants in the assembly as supplementary files.

**Reply to Reviewer 1:**

The manuscript on "Lake mixing regime selects methane oxidation kinetics of the methanotroph assemblage" by Mayr and Zimmermann et al, is a well written manuscript on the ecophysiology of methanotrophic bacteria.

Answer: Thank you for the positive assessment.

I only have the following remarks: Method section on the kinetics: - As you describe in detail how you eliminated outliers from the calculations I was wondering what the percentage of outliers was? From my (more marine) experience it is difficult to get good kinetic data, as many of my "Kinetic incubations" only gave erratic results:

Answer: We show the elimination (in %) in the revised manuscript and further clarified the procedure and its motivation (also with regards to a comment by reviewer 2). See section 2.3.

Filtering the samples water on the 0.2 ym filters. How long did it take to filter 1 - 2 liters on such a filter?? To my experience this may last very long: : ... Thus, did you use any prefilters? And could the RNA composition change of this presumably longer time??

Answer: We did not record the exact duration but in our experience, filtration takes typically less than 10 and always less than 15 minutes. In order to retrieve enough RNA for metatranscriptomics and at the same time reducing filtration time as much as possible we used large (142 mm) diameter filters.

We did not use prefiltration since some filamentous methanotrophs can be quite large, with a length of up to >100  $\mu$ m (Oswald et al., 2017). We took the samples with a niskin bottle and we connected the tubing for the filtration device directly to the niskin bottle. The sample was kept shaded (inside the niskin bottle) and did not experience major temperature changes in the cool autumn/winter weather of the sampling season for this experiment. Thus, changes from the in-situ transcriptional profile are expected to be minor. In-situ filtration might still be preferable, but the equipment for this was not available to us at the time of this work.

We added the approximate time and the diameter of the filter to the revised manuscript as well as more details to clarify the steps that were taken to insure rapid preservation of the transcriptome. See section 2.7.

Result /Discussion - Figure 1: I think it is essential also to show the methane concentrations, in situ MOXrates and also the cell numbers in one figure, as the "environmental background information".

Answer: Methane concentrations were already shown in Fig. 1. We changed numbers with arrows to horizontal lines with numbers and units to increase clarity. We only measured methane at the sampling depths in this study. We show cell numbers and methane oxidation rates in Supplementary Figure 1.

The second part of fig. 1 the kinetics should go in a separate figure, as this is more an experimental aspect.

Answer: We used this format deliberately to directly connect the core (kinetic) data of the study with the graphs showing the environmental situation, which we hoped would make it easier for readers to understand how these are linked. We would therefore prefer to keep this format.

- Comment on figure 2: to me another way of seeing the data is, that in the epilimnion Km is much higher in October, but is getting more and more similar to the hypolimnion.

Answer: There might be a misunderstanding here. The epilimnion data are shown in orange, and thus are actually lowest in October. (See Fig. legend)

Thus it is not a mixing of two compartments but more of an approximation of the epilimnetic traits to the hypolimnetic ones ??

Answer: Indeed, as we show in a work that has in the meantime been published in Communications Biology (Mayr et al., 2020)(https://doi.org/10.1038/s42003-020-0838-z) there is more going on than simple mixing, i.e. complex dynamics of the community, and species mixed in from the hypolimnion typically do not establish in the mixed layer. This was also reflected in the transcriptomes obtained for this work. By the end of the mixing process however, almost the entire water mass was indeed mixed, and we observed only a small remnant of the hypolimnion. We did briefly outline our understanding of the dynamics in section 3.2 (L. 296-305) and added one paragraph on the general dynamics in seasonally stratified lakes, from stratification to fully mixed water column. There we also define how we use "epilimnion" and "hypolimnion" in the context of this study (Section 3.1, L. 239-253).

**References:**

Mayr, M. J., Zimmermann, M., Dey, J., Brand, A., Wehrli, B. and Bürgmann, H.: Growth and rapid succession of methanotrophs effectively limit methane release during lake overturn, Commun. Biol., 3(1), 108, doi:10.1038/s42003-020-0838-z, 2020.

Oswald, K., Graf, J. S., Littmann, S., Tienken, D., Brand, A., Wehrli, B., Albertsen, M., Daims, H., Wagner, M., Kuypers, M. M. M., Schubert, C. J. and Milucka, J.: Crenothrix are major methane consumers in stratified lakes, ISME J., 11(9), 2124–2140, doi:10.1038/ismej.2017.77, 2017.

**Reply to Reviewer 2:**

Mayr, Zimmermann and colleagues studied the methane oxidation kinetics in the epiand hypolimnion of a eutrophic lake during autumn/winter lake overturn and report changing methane uptake kinetics. Likewise, changes in pmoC, A and B gene expression profiles were observed, indicating adjustments in the active methanotrophic community in dependence on methane availability. I see value in the presented work, but also limitations and open questions that would have to be clarified.

Answer: Thank you for your valuable comments. In the following we answered and clarified all open questions and limitations raised.

First of all, I request the authors to point out that they measured apparent methane oxidation kinetics. This should be clearly indicated throughout the manuscript.

Answer: We changed to "apparent methane oxidation kinetics" in the title and throughout the text.

In this study, all conclusions are derived from 1 - 2 l of water per sample, taken at C1 four different time points of the epilimnion and hypolimnion, respectively. However, no replicate samples were taken per layer. I find this hardly acceptable. How representative are the findings of 1 l water for the whole stratification layer of a lake? Can the authors be sure that the differences they see are indeed related to the respective water bodies? Already the molecular analysis of a second filter, which is a methodological replicate that was included for one sample, shows some differences (Fig. 3). Thus, I find it largely impossible to relate differences in the active methanotrophic community to stratification, especially in December and January and especially for pmoB and C (Fig. 3; statements 1. 266-267), without knowing anything about the biological variation within a layer. The repeated measurements over time provide some evidence, but do not solve this issue when it comes to minor differences between specific samples.

Answer: Horizontal mixing in lakes is strong (Lerman and Chou, 1995, p. 86) especially in stratified lakes and horizontal variation within a lake (excluding near-shore water or parts with limited water exchange) is expected to be small (Yannarell and Triplett, 2004) in comparison with the vertical variation during stratification and temporal variation of the mixed layer. Taking one profile, usually close to the deepest point is therefore common practice for studies in smaller lakes or lake surveys (Oswald et al., 2017). Sample volume per depth is then typically decided by the requirements of the analytical methods rather than from concerns about the volume's representativeness.

In our earlier study on Rotsee we investigated the general population structure around the oxycline (Mayr et al., 2020a) and later the community composition change over time during lake overturn in considerable detail. We found that especially the mixed surface layer is a very homogeneous environment considering both environmental conditions and the methanotroph assemblage (Mayr et al., 2020b). Therefore, we think that the epilimnion measurements are indeed representative for the mixed layer (epilimnion in this study) and only very small variabilities would be expected from multiple measurements. The focus of this study was not on variability within the mixed layer or within the hypolimnion, but rather to demonstrate that differences in kinetic parameters occur at all, in contrasting microbial communities within one lake which has not been demonstrated before. Therefore, although we did not study the variation within a layer in this study, we had considerable previous knowledge on the same lake from the years before and designed the study on this basis.

The hypolimnion during stratification by definition shows very different conditions to the epilimnion, but gradients of methane and other parameters can be observed within the hypolimnion. Therefore, whereas the hypolimnion is less homogeneous than the epilimnion, the difference to the epilimnion is clear, based on both, the environmental conditions (orders of magnitude higher methane concentrations) and a different methanotrophic assemblage (this study and Mayr et al., 2020b, 2020a).

Regarding the request for (biological) replication of the kinetic measurements, we have to note that, although measuring different depths and or replicates from the layers would be favourable, we had to adjust the amount of incubations to our handling limit of the required incubations. We decided to improve the replication for the respective depths to have enough data points to determine the affinity of the respective depth (see our answer on outlier elimination to the other reviewer). Thus, in order to have enough different methane concentrations and technical replication we could only analyse two depths. This decision was also based on the results obtained in our previous study of the lake overturn period as discussed above (Mayr et al., 2020b). Therefore, we could not have analysed more replicates or depths on one date due to practical limitations on the number of incubations that we could handle, but we believe that the multiple time points provide confidence in our measurement. The reviewer may note that the existing studies on apparent MO kinetics in lakes were indeed all done on lower numbers of samples.

We agree that the difference between epi- and hypolimnion become smaller in December and January, which is expected as the epilimnion is increasing in size and stratification of the lake weakens. In January the hypolimnion is almost gone and only small differences remain between epi- and hypolimnion, both in terms of methane concentration and methanotroph composition. We agree that the characteristics of the epi- and hypolimnion change over time (which is expected and was part of our research question) and use the terms epi- and hypolimnion more to clarify where the sample was taken, rather than to imply homogeneity within the category. Because there is a temporal variation, we also do not give average values for the epi- or hypolimnion. We agree that the term hypolimnion for the bottom sample in January might be misleading without further information, because the lake has almost completely mixed, and the strong difference seen between epi- and hypolimnion before are no longer present.

In the revised manuscript we added one paragraph on the general dynamics in seasonally stratified lakes (L. 239ff), from stratification, lake mixing and the fully mixed water column to section 3.1. Further, we explain within this paragraph and the following the characteristics of epi- and hypolimnion and how we use "epilimnion" and "hypolimnion" in the context of this study, and include some information on our pervious study (Mayr et al., 2020b; Zimmermann et al., 2019) on which we based the present study.

Besides, the reason for including one experimental replicate (January, hypolimnion) or the conclusions derived from this sample are not mentioned anywhere.

Answer: We included this replication in order to increase confidence in the transcriptomic analysis. It is not easy to obtain enough RNA for metatranscriptomics from our lake samples, which we tried to optimize as much as possible considering the trade-off between 1) filtration time and 2) obtaining enough sample for RNA extraction and subsequent sequencing. Because both effort and costs are high, we included only one replicate, although we of course agree more would be favourable. Typically, RNA-based measurements show a somewhat larger variability than DNA-based sequencing results, therefore we think that our replicate shows that the results are very reproducible for a metatranscriptomic analysis, but the result also cautions that small differences between samples should not be overemphasized. That all three January samples are basically indistinguishable based on metatranscriptomics is not surprising as the mixing process is almost complete in January and the conditions at the sampled depths are very similar at this date (see answer above).

We added a short explanation to the revised manuscript (Methods section 2.7, L. 184) and also added conclusions derived from this sample to the revised manuscript (Results and Discussion, section 3.3, L. 412).

How do the authors know that they had a representative sample from the hypolimnion at the last sampling date? There is no change in temperature evident and the decline in oxygen concentrations does not reach oxygen concentrations as low as at the earlier time points. Likewise, methane concentration in this sample is not as high as in the other samples from the hypolimnion. Thus, it appears that the sample was not taken at appropriate depth to be comparable with the others.

Answer: During lake overturn the mixed layer (corresponding to epilimnion in our study) increases over time due to gradual cooling and the resulting density change. By January this mixing process is almost complete and the conditions in the remaining "hypolimnion" differ from those earlier in the season. However, the water at the lake bottom at the time of sampling still showed a small differences of e.g. methane and oxygen concentration to the water above, although the temperature gradient is gone (Figure 1d). The lake has a max. depth of 16 m and our sample was taken from 15 m, the maximum depth that can be sampled with the niskin bottle without disturbing the sediment. Therefore, if the sample in January is truly a "hypolimnion" sample is indeed somewhat debatable but refers to the bottom location of the sample as for the other sampling dates.

In the revised manuscript we added one paragraph on the general dynamics in seasonally stratified lakes and defined our use of epi- and hypolimnion more clearly (see answer above).

The conclusion about the specific enrichment of well-adapted methanotrophs with particular methane oxidation kinetics (l. 23) is conceivable, but should be drawn more carefully, because it remains unclear whether the observed kinetics are indeed adaptations of particular competitive methanotrophs under oligotrophic conditions, especially with regard to affinity. As only apparent parameters could be estimated, it remains unclear whether the methane monooxygenase of the respective organisms has indeed a higher affinity (lower Km) and is thus more competitive. It should be kept in mind in this context that a low apparent Km is not necessarily a specific adaptation to low methane concentrations, but can be the result of starvation (see Dunfield and Conrad 2000, AEM).

Answer: It is of course true that we only measured the apparent kinetic properties of the whole MOB assemblage. Our conclusion is based on the observation that changes in the apparent kinetic properties of the assemblage are accompanied by changes in the community composition and changes in the expression level of *pmoCAB* variants of the assemblage. In this respect, we draw conclusions on an average trend on the community level but we are well aware that there might be additional mechanisms, e.g. on individual species level as well.

In the revised manuscript we stated (L. 24) more carefully *as one* important factor for creating niches. Further, we discussed (L. 307-311 Results and Discussion section) the additional explanation, that changes in  $K_m$  can be a result of starvation (Dunfield and Conrad, 2000). We also note that Dunfield and Conrad (2000) observed a constant specific affinity (a0), which is in contrast to our results. In our case both  $K_m$  and a0 change with depth and with time, which suggests adaptation rather than starvation as described in Dunfield and Conrad (2000).

Whether the kinetic differences, assuming they have a genetic basis, have an influence on the competitiveness of a species is indeed a very interesting question. We are currently preparing a manuscript that discusses a modelling approach in which we investigate the interplay of MOB populations consisting of "species" with varying Km and Vmax and their competitiveness in the Rotsee setting in detail.

To determine methane uptake kinetics (Fig. 2), the samples were apparently incubated at the temperatures measured in the epilimnion. However, samples from the hypolimnion encounter much lower temperatures in autumn. How does that affect comparability of the obtained results and conclusions about in situ conditions? This should be taken into account.

Answer: We clarified the rationale for our approach in the methods section 2.3 (L. 118-120) and mention the caveats that arise from this choice. Our approach was not designed to obtain the most realistic in-situ rates, but to get a valid comparison of the apparent kinetics in the different communities of the sampled water layers. To be able to compare the measured kinetics between epi- and hypolimnion at each given date, we measured them at the same temperature, since temperature influences the solubility of gases (methane).

Related to this point: Considering that altered temperature and oxygen conditions were used to characterize the methane uptake kinetics in vitro, to what extent can the findings be translated to in situ conditions, considering that these factors can affect the measured Km and Vmax (see the study of Thottathil et al 2019, who report that increasing oxygen concentrations in lake water can reduce maximum methane oxidation rates;

doi.org/10.1007/s10533-019-00552-x). Is it conceivable that Vmax in the hypolimnion is underestimated when determining oxidation rates at higher oxygen concentrations in vitro?

Answer: Our main goal was to compare apparent kinetics as a trait of microbial communities rather than to obtain precise information on in-situ kinetics. In an attempt to measure methane affinities under as standardized conditions as possible we measured the methane oxidation kinetics under similar conditions to the best of our knowledge. We added this rationale behind the methane affinity experiments to the methods section. Further, we add that altered temperature and oxygen concentration as compared to the in-situ conditions may influence the result as a limitation of our study to the discussion section (L. 291-295).

I find it very unfortunate that the identification of methanotrophs stops at the level "type Ia, type Ib, type II". The sequence information should provide more detailed information about the identity of the methanotrophs. At least for pmoA comprehensive datasets are available covering besides cultivated strains diverse groups of uncultivated taxa, so that more information could have been extracted here to identify conspicuous taxa.

Answer: In the revised manuscript we provided an improved classification of *pmoA* based on the NCBI refseq\_protein database and a *pmoA*-based phylogenetic tree (Supplementary Figure 2) of the sequences found in this study and reference sequences. The taxonomic affiliations were also added to Figure 3 and Supplementary Figure 3. We note that the *pmoABC* sequences are available as fasta files via our data repository.

Specific comments: l. 19 and 291: According to the data in table S1, the difference in Km is 20-fold, not 2 orders of magnitude

Answer: That's correct, we have corrected this.

1. 25: Where in the presented work is it shown or discussed that 90% of the methane are removed? It appears that this is not a conclusion that is derived from the presented work.

Answer: This is a result from our work on the overturn of the same lake one year prior to the present study (Zimmermann et al., 2019), and is based on using mass balance analysis and modelling. We provided a clear reference for the statement in question in the revision (L. 41-44) and rephrased the sentence in the Abstract.

1. 65: Metagenomic data were used as a basis for the metatranscriptomic data analysis, but are not presented independently; thus, I would not emphasize the metagenomics approach here for the analysis of MOB assemblages.

Answer: We deemphasized the metagenomics part here.

1. 73: Five campaigns in autumn 2017 does not appear correct (three samplings in 2017 and one in 2018 according to the presented results)

Answer: Yes, we have changed this accordingly.

1. 74-75: More measured parameters are given here than presented; harmonize.

Answer: We harmonized the methods part accordingly in the revised manuscript.

1. 78: I do not find any helpful information about the radio isotope tracer technique in Steinle et al 2015. While the cited references enabled me to understand how methane oxidation rates were determined, they do not allow me to evaluate whether/how this procedure can be used to survey methane oxidation kinetics.

Answer: We improved the methods section and added reference to Lofton et al. (2014) who derived methane oxidation kinetics from methane oxidation rates that were measured using  ${}^{14}$ C-CH4. As described in Bussmann et al. (2015), the  ${}^{3}$ H-CH4 tracer technique is more sensitive than the  ${}^{14}$ C-CH4 technique and therefore allows shorter incubation times and rate determination at low CH4 concentrations. The combination of the approaches is a novelty of our work.

1. 80: How much methane was in this mixture?

Answer: We specified the amount of methane in the mixture in the methods section. L. 93-95 "The specific activity of 3H-CH4 is 0.74 TBq mmol-1 and the 200  $\mu$ L of gaseous 3H-CH4/N2 mixture therefore contained 108 pmol 3H-CH4. In comparison, the 500  $\mu$ L gas bubble with the lowest concentration of unlabelled methane, contained 17 nmol CH4."

1. 100-106: The authors describe different criteria that were used to identify and eliminate outliers here. Point four states that data points were removed in case less then two replicates remained. According to 1. 93, duplicates were prepared. Does that mean that data for a specific methane concentration were lost each time one of the two replicates was identified as outlier? In this context, it is also unclear what Fig. 1 e-h shows. Do the presented data points represent individual measurements or are these mean values of the two replicates? Sometimes, I see two data points at a specific concentration, but sometimes I see only one point. Please clarify.

Answer: In a revised manuscript we have improved clarity and the level of detail in the methods section 2.,3 L127+.

It would be valuable to know how many high-quality reads the authors generated per sample in the metagenomic and metatranscriptomic analysis, respectively.

Answer: For metagenomics we generated approx. 31-37 million reads and for metatranscriptomics 26.6 - 34 million reads. We provided this information in the methods section of the revised manuscript. Section 2.7, paragraph 2.

1. 157: Why three samples in October; to my understanding there should be one from the epilimnion and one from the hypolimnion per point of time.

Answer: We measured one additional depth in October in between epi- and hypolimnion, which is included in the data repository but is currently not discussed in the manuscript. We did not pursue the intermediate sample in later campaigns since we concluded that focusing our effort on the Epilimnion and Hypolimnion (e.g. continuing with measurement triplicates) would be better. We did however use the metagenomics data from this sample to improve the assembly. We stick to the two-depth presentation for the main manuscript and figures as it keeps the story focused. However, we added the information on this additional sample in the methods section (Metagenomic and metatranscriptome analysis, 2.7, L. 203-205).

1. 163: Can a few words be added to describe this custom database? How was it set up? What type of data does it include?

Answer: We prepared three custom databases, one for each gene: pmoA-like, pmoB-like, pmoC-like. We extracted *pxmABC*, *pmoCAB2*, *pmoCAB* from both alpha- and gammproteobacterial genomes manually. Note that *pmoCAB2* is only known from alphaproteobacterial methanotrophs as yet. We used amino acid sequences with diamond blastp to find the genes in our metagenome and transcriptome. The custom database was especially important for *pmoC* which was not included in the available prokka databases. A short description was added to the revised manuscript (section 2.7, L. 213-215), and the databases were added as supplementary files 1-3.

1. 202-205, 1. 295 and perhaps elsewhere: wording: do the authors refer to Km or a0 here when talking about affinity?

Answer: We refer to  $1/K_m$  when talking about *affinity* (L. 106). We refer to  $\mathbf{a}^\circ$  (L. 141) when we talk about *specific affinity* (defined as the ratio  $V_{max}/K_M$ .). We made sure that these terms are clearly defined on first use and put the kinetic parameter terms in italics throughout to help to distinguish them in the text.

1. 204-207: I cannot follow argumentation here. And how do the authors explain that the organisms with the higher Vmax and lower Km disappear in January (Fig. 1h), although they should have a competitive advantage?

Answer: The fact that certain types disappear when they should have a competitive advantage with respect to the methane oxidation kinetics (at least insofar as we can conclude on the traits of individual species from the measurements performed on consortia) indeed leads us to the conclusion that one would need to consider additional traits to explain the observed abundance pattern and its dynamics. As stated on L. 290-302 we believe that there have to be additional important differences in other traits (i.e. temperature optimum or specific adaptations to low or high oxygen concentrations). We already have strong indications from our previous work that these factors are indeed important (Mayr et al., 2020a, 2020b). Nonetheless, we do observe significant differences and changes in the methane oxidation kinetics which are related to the differences and changes in the environment. We have incorporated the information above in sections 3.2 (L301-305) and 3.3 (L. 407-4017) of the revised manuscript.

l. 241: A range of 1 - 40 is a bit outdated. Atmospheric methane oxidizers in soil are meanwhile known to have a0s values with up to 195 x 10-12 L/cell\*h (Tveit et al) and in upland soils, estimates are ranging up to 800 x 10-12 L/cell\*h (Kolb et al 2005; doi:10.1111/j.1462-2920.2005.00791.x)

**Answer: Thank you, we updated this.**

1. 248-250: I find the 25% and 93% values critical here, because huge differences are observed at the individual time points. Especially the 93% value appears to be strongly affected by the huge difference observed in December.

Answer: We have now formulated a more differentiated argument: Individual percentages can be calculated from the individual values for the maximum methane oxidation rate as well as the in-situ methane oxidation rate in Supplementary Table 1. In the epilimnion there might indeed be a trend over time, however all values are below 50%. In the hypolimnion we observe less variation and all values are above 67%. Therefore, we do believe that it is a valid conclusion that MOB in the epilimnion were generally more limited than MOB in the hypolimnion.

1. 254: What do the authors mean with aggregate properties here? What aggregates do they refer to?

Answer: "Thus, the reported kinetics reflect aggregate properties of the respective assemblage." Aggregate was used in the sense of "formed or calculated by the combination of several separate elements", in this case of all the species and individuals in the community, i.e. we meant to express that the reported kinetics reflect the (apparent) properties of the MOB assemblage which is present at the depth and date of sampling. In the revised manuscript we change the term to "composite".

1. 256 – 258: It would be very valuable if the described findings could be seen in Figure 3.

Answer: We rearranged Figure 3 according to the remarks of reviewers 2 and 3 to better visualize the changes in the transcriptionally active methanotroph assemblage.

We calculated a correspondence analysis (cca, vegan) instead of the detrended correspondence analysis (decorana, vegan) (detrending was not necessary, because no arch effect was visible) because the more common scaling option 2 is not implemented in decorana. Further, we made the taxonomic affiliations visible with a color code in

the ordination (Fig. 3a) and added taxonomic information to the legend for Fig. 3b. To save space and because similar information is shown in the analogous figures for *pmoB* and *pmoC*, we moved them to the Supplementary Material (Supplementary Fig. 3a-d) including taxonomic information in the legend.

1. 292-293: The transcription of genes does not relate to enzyme affinity or apparent Km values; thus, I cannot follow argumentation here.

Answer: The association is only correlational so far, this is true. We clearly stated this as a hypothesis in the revised manuscript (L. 429).

1. 301: I do not necessarily agree to the term "entirely" in the context with "kinetic traits"; other environmental conditions may have affected the kinetic parameters. Please keep in mind that you can only measure apparent parameters, not enzyme kinetics.

Answer: We have removed the "entirely" there (L. 440). Yes, it is true that other parameters likely influence kinetic traits. We keep in mind that we do not measure enzyme kinetics but apparent kinetics. Nevertheless, our metatranscriptomics data clearly suggest that pMMO was by far the most expressed methane oxidizing enzyme. Therefore, according to our data most of the methane which is oxidized is oxidized by pMMO and not other enzymes.

1. 303-304: Please note that Methylocapsa gorgona does not possess a second pmoA gene for "high-affinity oxidation" despite being able to live on very low methane concentrations (Tveit et al).

Answer: Yes, we are aware of this interesting study and cite it elsewhere in the manuscript. *Methylocapsa gorgona* can live and grow on very low methane concentrations, but it does not have a very low apparent Km (4.9  $\mu$ M), but it has a very high specific affinity. But many methanotrophs possess sMMO and pMMO, and sMMO has a lower affinity than pMMO. To our knowledge *Methylocapsa gorgona* does not have sMMO.

References: The reference list does not allow to differentiate publications (e.g. Mayr et al 2019a, b, c). The reference list lacks information about the year the work has been published and the indices a,b,c.

Answer: Here we followed the author guidelines of the Journal and added the year + a,b,c at the end of the reference. The year changed in two cases.

Mayr, M. J., Zimmermann, M., Dey, J., Wehrli, B. and Bürgmann, H.: Data for: Community methane-oxidation kinetics selected by lake mixing regime [Data set], Eawag Swiss Fed. Inst. Aquat. Sci. Technol., doi:10.25678/0001fa, **2019**.

Mayr, M. J., Zimmermann, M., Guggenheim, C., Brand, A. and Bürgmann, H.: Niche partitioning of methaneoxidizing bacteria along the oxygen-methane counter gradient of stratified lakes, ISME J., 14(1), 274–287, doi:10.1038/s41396-019-0515-8, **2020a**.

Mayr, M. J., Zimmermann, M., Dey, J., Brand, A., Wehrli, B. and Bürgmann, H.: Growth and rapid succession of methanotrophs effectively limit methane release during lake overturn, Commun. Biol., 3(1), 108, doi:10.1038/s42003-020-0838-z, **2020b**.

Figure 1: The axis showing oxygen concentrations should have a more increments.

Answer: We provide more increments on the oxygen axes in the revised Fig. 1.

Figure 2: explain error bars

Answer: We improved clarity of the caption of Fig. 2.

Figure 3: The distinction by color is difficult in plots a1-c1; why not choosing more distinct colors / a broader range of colors per plot? This is of particular importance, as the relative abundances cannot be taken from Table S2 without additional calculations. It is currently impossible to identify type Ib or type II methanotrophs based on the color code and without further invest. However, as pointed out above, it would be even more valuable if more taxonomic information could be provided.

Answer: Thanks for the valuable suggestions for Figure 3. Unfortunately, we could not find a better and still colour-blind proof color palette. We decided that taking a clear brightness-based palette with additional colour information would be best. As described above we made general improvements to the data visualization of Fig. 3 based on the comments of reviewer 2 and 3 by improving the ordination-visualization and moving it to the main text. Further, we added improved taxonomic information to the legend and provide a *pmoA* phylogenetic tree in the supplementary material. Further, we provide the supplementary table with the underlying data as a tab delimited file instead of a pdf to make it more accessible. The fasta files of all *pmoABC* variants are available in the provided data repositories ("Data availability").

Table S3: Provide reference for Knief et al 2015.

Answer: We provided the reference in the revised manuscript.

**Reply to Reviewer 3:**

This manuscript describes the study of methane oxidation (MOX) during lake overturn in Lake Rotsee in Switzerland. You combine measurements of MOX kinetics with metatranscriptomic analyses of methane monooxygenase genes and report differences between epi-and hypolimnion during stratification and a convergence of MOX kinetics and gene expression during lake water mixing. You conclude that methane oxidizers with well-adapted kinetics occupy distinct niches in stratified lakes.

Answer: Thank you for your valuable comments. In the following, we address all open questions and limitations raised.

While I think the report of kinetic parameter of methane oxidation is of great relevance, however, I found that the manuscript suffers from a lack of clarity and over-simplifications. Most importantly, it's unclear how the central conclusion, that welladapted methanotrophs inhabit niches depending on methane availability (in hypo- and epilimnion), is reached. Wouldn't a match between in situ CH4 concentration and Km (not normalized per cell) be a stronger indication of such an adaptation?

Answer: Thank you for your positive assessment of the relevance of reporting kinetic parameters. We improved the clarity of our central conclusion and added a steady state calculation to show how Km values and in-situ methane concentrations are linked to the supplementary material "Supplementary Calculation". Further we discuss this calculation in the main text in L. 329-338:

"We thus concluded that MOB assemblages displayed a specific adaptation to the prevailing methane concentrations based on the fact that we observed a higher *affinity* (low  $K_m$ ) in the low-methane epilimnion compared to the methane rich hypolimnion as long as stratification is present. That the  $K_m$  values of the assemblage in the epilimnion do not match the *in-situ* methane concentrations is not unexpected: In the mixed layer and under the assumption of a steady-state, the flux of methane from the hypolimnion is balanced by the methane oxidation rate. Under these conditions, the *in-situ* methane concentration depends on the half-saturation constant ( $K_m$ ) but should be lower than it (Supplementary Calculation 1). Per definition, the half-saturation constant is the substrate concentration where the growth rate is half the maximum growth rate. Even if the growth rate is only half the maximum growth rate, microbial methane oxidation continues, and methane concentrations decrease to values below  $K_m$ ."

I also believe a better use of the metatranscriptomic data could help to strengthen this point. A finer taxonomic resolution based on the pmoCAB genes and a more quantitative characterization of the community turnover should be possible – and could help to make the point that indeed there are distinct populations of MOB that are adapted to in situ CH4 concentration. Accordingly, I think that Figure 3 a1-c1 is not the ideal way to convey this important point. Maybe a combination of SI Fig. 2 (which I think shows quite nicely the convergence towards similar gene expression patterns in January, with a Figure showing the taxonomic composition of MOB during lake overturn would be a better choice.

**Answer: Yes, we agree.**

We provided a finer taxonomic resolution for *pmoA* in the revised manuscript. We now provide a phylogenetic tree in Supplementary Figure 2 and added this classification to Figure 3a and b, as well as Supplementary Figure 3a, b. *pmoC* and *pmoB* are less indicative for taxonomic classification because they are not as frequently sequenced and used for this purpose. Therefore, we only classified them to family level and used the same finer taxonomic resolution if a respective sequence was found on the same contig with a taxonomically classified *pmoA*. The full list is provided as .txt file (Supplementary file 4). We note that the *pmoABC* sequences are available as fasta files via our data repository.

Thanks for the suggestion regarding Figure 3. We rearranged Figure 3 according to the remarks of Reviewer 2 and 3 to better visualize the changes in the transcriptionally active methanotroph assemblage.

We calculated a correspondence analysis (cca, vegan) instead of the detrended correspondence analysis (decorana, vegan) (detrending was not necessary, because no arch effect was visible) because the more common scaling option 2 is not implemented in decorana. Further, we made the taxonomic affiliations visible with a color code in the ordination (Fig. 3a) and added taxonomic information to the legend for Fig. 3b. To save space and because similar information is shown in the analogous figures for *pmoB* and *pmoC*, we moved them to the Supplementary Material (Supplementary Fig. 3a-d) including taxonomic information in the legend.

Moreover, I was somewhat irritated by the rather vague description of the environmental conditions during lake overturn. The traditional definition of lake stratification and hence the difference between epi- and hypolimnion based on temperature rather than oxygen. And while the manuscript addresses MOX during lake overturn, you refer to the oxycline for sampling. I understand that the temperature profiles shown in SI Fig 1 may not be as clear as the oxygen profiles shown in Fig. 1 - but I would advise to show all profiles (also conductivity which should explain the inverse stratification pattern in December) and to be very clear with the definition of overturn, thermosand oxycline.

Answer : We added temperature profiles to Figure 1 and used the terms overturn, thermo- and oxycline more carefully.

Finally, given the relatively low number of samples and the fact that the pattern was (only) observed in Lake Rotsee, I think the manuscript should be thoroughly rewritten to make clear that this may reflect a specific situation in the (relatively eutrophic) Lake Rotsee. Also, there are several cases of speculation or exaggerated extrapolation, which should be avoided.

Answer: The effort of obtaining the time resolved data for two water layers was very substantial (described in the answers to Reviewer 1 and 2). Nevertheless, we agree that we only have a limited number of samples from a single lake. We highlighted this more clearly and explicitly stated that further investigations in other lakes will be required to confirm our findings.

L 11 In freshwater lakes... so, this excludes saline lakes? Consider removing "freshwater"

Answer: We removed freshwater here and mention it in Line 14: "in a seasonally stratified freshwater lake".

L 14 we tested the hypothesis that methanotroph assemblages in a seasonally stratified lake...

Answer: We changed this accordingly and toned down the conclusions in order to avoid overselling.

L 18 consider a brief explanation of the meaning "half-saturation constant" here

Answer: We added a brief definition of "half-saturation constant" (L. 19-20).

L19 ...Km differed by two orders of magnitude – but in the results it seems that they differed between 15 and 0.7 uM (a factor of  $\sim$ 20)

Answer: We corrected this.

L 25 ...90% of what?

Answer: 90% of the methane that is transferred to the epilimnion during lake overturn is oxidized. Since this is a result from an earlier study we rephrased the sentence (L.43-44).

L28 can you talk about a climate IMPACT of lacustrine systems?

Answer: Yes, we think so. According to DelSontro et al. (2018), lacustrine systems emit an equivalent of about 20% of the global fossil fuel  $CO_2$  emissions, and methane contributes approx. 75% to this. Continuing eutrophication of lacustrine systems will most likely further increase emissions by another 30-90% (Beaulieu et al., 2019; DelSontro et al., 2018). We clarified: "Methane is a major contributor to the climate impact of the greenhouse gas emissions from lakes".

L 31 anoxic habitats.... In the oxygen-depleted hypolimnion... repetitive

Answer: We removed "anoxic habitats" in the revised manuscript.

L 47 kinetic traits... Use kinetic parameter instead (see L 48)

Answer MZ: We have harmonized the use of "kinetic traits" and "kinetic parameter" in the revised manuscript.

L 58 ... Lake Rotsee...

Answer: We removed the "a" written before "Lake Rotsee".

L 63 ex situ consider replacing with "laboratory incubations"

Answer: We replaced this as proposed (L. 71)

L 73 four or five campaigns?

Answer: We corrected this to four campaigns.

L 77 and onward. Please provide more detail on this method including how the killed controls were treated.

Answer: We provided more detail on how the killed controls were treated (section 2.3, L.110-116).

L 91 how were Schott bottles sealed air-tight?

Answer: The Schott bottles were not sealed air-tight. Since gasses were stripped from the samples anyway for these analyses, this was not necessary. For determinations of methane concentration and "in-situ" rates, we directly filled samples into serum vials and those were sealed air-tight (see sections 2.4, 2.5).

L 110 we determined the in-situ MOX rate... in duplicate ex-situ incubations... Confusing, please rewrite.

Answer: We removed "in-situ" and changed ex-situ incubations to laboratory incubations throughout the manuscript.

L 161 an 167 reads shorter than 400 or 300 bp were removed?

Answer: We first used a general approach to identify genes (prodigal) removing <400bp genes. During targeted gene identification with prokka and diamond again shorter gene pieces of MMO were identified and we removed short gene fragments <300bp in the original manuscript. In the revised manuscript we harmonized this and used 400bp base pairs as a cut-off in both steps.

L 183 aerobic methane oxidation likely contributed to this oxygen depletion in the epilimnion. This seems very speculative for me. Could a back of the envelope calculation, e.g. knowing the volume and CH4 concentration in the hypolimnion and the stoichiometry of MOX be used to support this speculation?

Answer: We added a short calculation to the main text that supports that methane oxidation substantially contributed to the oxygen depletion. (Section 3.1, L.261-278)

"The oxygen concentration shifted from 15% oversaturation in October to 67% undersaturation in December (Fig. 1a-d), aerobic methane oxidation likely contributed to the oxygen depletion in the epilimnion, which we substantiate with the following calculation: The stoichiometry of microbial methane oxidation is:

 $CH_4 + (2 - y)O_2 \rightarrow (1 - y)CO_2 + yCH_2O^{BM} + (2 - y)H_2O$ where y is the carbon use efficiency and  $CH_2O^{BM}$  designates MOB biomass. Based on theoretical considerations and experimental data, a carbon use efficiency of 0.4 has been reported (Leak and Dalton, 1986). This means that per mole of methane 1.6 moles of oxygen are used. The mixed layer depths for the four sampling campaigns are roughly 6, 10, 12 and 14 m, corresponding to mixed layer volumes of 2.5, 3.7, 4.1 and 4.3 GL in Lake Rotsee. Multiplying the measured methane oxidation rates in the epilimnion with these volumes results in a total methane oxidation of 600, 11560, 11800 and 200 mol d-1. Integrated over the time period of the four campaigns, this results in a total of 0.66 Mmol of methane that were oxidized with 1.1 Mmol of oxygen from the epilimnion. In an average volume of the mixed layer of 3.7 GL with an initial concentration of 340  $\mu$ M (10.9 mg L-1) of oxygen, this would reduce the oxygen concentration by 180  $\mu$ M to 160  $\mu$ M or to about 5 mg L-1. Note that possible oxygen production and exchange with the atmosphere, as well as additional oxygen sinks are not included in these considerations."

L 228 critical phase - critical for what?

Answer: We clarified that this phase is critical for potential outgassing to the atmosphere and thus for climate relevance. Section 3.4, L. 342.

L 233 specific affinity towards methane... unclear what is meant here.

Answer: We rephrased: "specific methane affinity".

L 235 was the convergence only driven by changes in kinetic parameter in the epilimnion (or also in the hypolimnion as seems apparent from Fig. 2 a)

Answer: We clarified that the convergence was driven by changes in kinetic parameter in the epilimnion and the hypolimnion. L.351-356.

L 289 remove "as in many other stratified lakes" – too speculative (or include references, but I would not advise so in the conclusion part)

Answer: We provided references in the revised manuscript (L. 427).

That low methane concentrations are found in the epilimnion and higher methane concentrations in the hypolimnion is very common, especially in small lakes e.g. (Bastviken et al., 2004; Borrel et al., 2011), which can vary seasonally like in Rotsee, or in permanently stratified lakes elevated methane concentrations can persist over many decades. Continuing eutrophication of lakes and lack of recovery of eutrophic lakes will likely increase the number of lakes with anoxic methane-rich bottom waters in future (Beaulieu et al., 2019; Jenny et al., 2016).

We toned down the generalized claims at the end of the conclusion in the revised mansucript in order to put our results into context without overstating generalizability.

L 295 adaptation to oligotrophic conditions - Lake Rotsee can not be considered oligotrophic

Answer: What we mean is adaptation to low methane availability, we removed the term accordingly (L. 434).

L 298 transport of methane into the epilimnion provided and advantage for fast-growing

MOB over slower competitors. This is not shown (at least in this manuscript) and should be removed.

Answer: We have provided evidence for the dynamic adaptation in (Mayr et al., 2020), but the reviewer is correct that the phrasing here is misleading and while our data here is in line with the previous investigation the conclusion cannot be made from the data in this paper. We have rephrased this sentence in the revised manuscript and removed the respective part (L.436).

 \* These authors contributed equally to this work.

[revised manuscript text omitted]